# Mechanistic insight into TRIP13-catalyzed Mad2 structural transition and spindle checkpoint silencing

Melissa L. Brulotte [1], Byung-Cheon Jeong[1], Faxiang Li[1], Bing Li[1,2], Eric B. Yu[1], Qiong Wu[3], Chad A. Brautigam[3,4], Hongtao Yu [1,2] & Xuelian Luo[1,3]

The spindle checkpoint maintains genomic stability and prevents aneuploidy. Unattached kinetochores convert the latent open conformer of the checkpoint protein Mad2 (O-Mad2) to the active closed conformer (C-Mad2), bound to Cdc20. C-Mad2–Cdc20 is incorporated into the mitotic checkpoint complex (MCC), which inhibits the anaphase-promoting complex/cyclosome (APC/C). The C-Mad2-binding protein p31[comet] and the ATPase TRIP13 promote MCC disassembly and checkpoint silencing. Here, using nuclear magnetic resonance (NMR) spectroscopy, we show that TRIP13 and p31[comet] catalyze the conversion of C-Mad2 to O-Mad2, without disrupting its stably folded core. We determine the crystal structure of human TRIP13, and identify functional TRIP13 residues that mediate p31[comet]–Mad2 binding and couple ATP hydrolysis to local unfolding of Mad2. TRIP13 and p31[comet] prevent APC/C inhibition by MCC components, but cannot reactivate APC/C already bound to MCC. Therefore, TRIP13–p31[comet] intercepts and disassembles free MCC not bound to APC/C through mediating the local unfolding of the Mad2 C-terminal region.

[1] Department of Pharmacology, University of Texas Southwestern Medical Center, 6001 Forest Park Road, Dallas, TX 75390, USA. [2] Howard Hughes Medical Institute, University of Texas Southwestern Medical Center, 6001 Forest Park Road, Dallas, TX 75390, USA. [3] Department of Biophysics, University of Texas Southwestern Medical Center, 6001 Forest Park Road, Dallas, TX 75390, USA. [4] Department of Microbiology, University of Texas Southwestern Medical Center, 6001 Forest Park Road, Dallas, TX 75390, USA. Correspondence and requests for materials should be addressed to H.Y. (email: Hongtao.Yu@utsouthwestern.edu) or to X.L. (email: Xuelian.Luo@utsouthwestern.edu)

The spindle checkpoint is a cell-cycle surveillance system that ensures proper chromosome segregation in mitosis[1–3]. Activated by unattached kinetochores, checkpoint proteins collaborate to inhibit the anaphase-promoting complex or cyclosome bound to its mitotic activator Cdc20 (APC/C$^{Cdc20}$)[4–7] until all chromosomes achieve bipolar attachment to spindle microtubules. A major effector of the spindle checkpoint is the mitotic checkpoint complex (MCC), which consists of BubR1–Bub3, Cdc20, and Mad2. MCC binds to the substrate-binding site of APC/C$^{Cdc20}$, preventing it from recognizing and ubiquitinating securin and cyclin B1. Stabilization of securin and cyclin B1 delays sister-chromatid separation and mitotic exit.

A rate-limiting step in MCC formation is the conformational activation of the checkpoint protein Mad2[8, 9]. Mad2 is an unusual protein with multiple folded conformers, including the latent open conformer (O-Mad2) and the active closed conformer (C-Mad2)[10–14] (Fig. 1a). C-Mad2 can bind to Mad1, Cdc20, and other proteins through the Mad2-interaction motif (MIM)[4, 11, 12, 15]. O-Mad2 and C-Mad2 share the same stably folded core. They differ in their N- and C-terminal regions. When bound to a ligand, the C-terminal region of C-Mad2 wraps around the MIM and stretches across the core like a "safety belt" (Fig. 1a, right panel). Mad2 forms a constitutive complex with Mad1, in which Mad2 adopts the C-Mad2 conformation[12]. The C-terminal region

of O-Mad2 does not form the safety belt and blocks the ligand-binding site (Fig. 1a, left panel). Because C-Mad2 is thermodynamically more stable than O-Mad2, O-Mad2 alone, in the absence of a ligand, can spontaneously convert to the unliganded C-Mad2 conformation, in which the safety belt is formed but does not trap a ligand[13, 16] (Fig. 1a, middle panel).

Unattached kinetochores promote Mad2 activation and MCC formation through a series of signaling and recruitment events[1–3]. The KMN network of proteins (consisting of Knl1–Zwint, the Ndc80 complex, and the Mis12 complex) serves both as a major microtubule receptor and as a docking platform for checkpoint proteins at outer kinetochores, and is required for spindle checkpoint signaling[17]. When kinetochores are not attached by microtubules, the Ndc80 complex (Ndc80C) is phosphorylated by Aurora B and recruits the checkpoint kinase Mps1[18, 19]. Mps1 phosphorylates Knl1, providing docking sites for the Bub1–Bub3 complex[20–27]. Bub1–Bub3 phosphorylated by Mps1 and Cdk1 further recruits the Mad1–C-Mad2 complex, and other MCC components BubR1–Bub3 and Cdc20[28–36]. The Mad1–C-Mad2 complex at kinetochores recruits additional O-Mad2 from the cytosol and converts it to an intermediate form (I-Mad2)[14, 37, 38]. Through a high-energy transition state in which its C-terminal region is locally unfolded, I-Mad2 binds to Cdc20 to form the Cdc20–C-Mad2 complex, which then associates with BubR1–Bub3

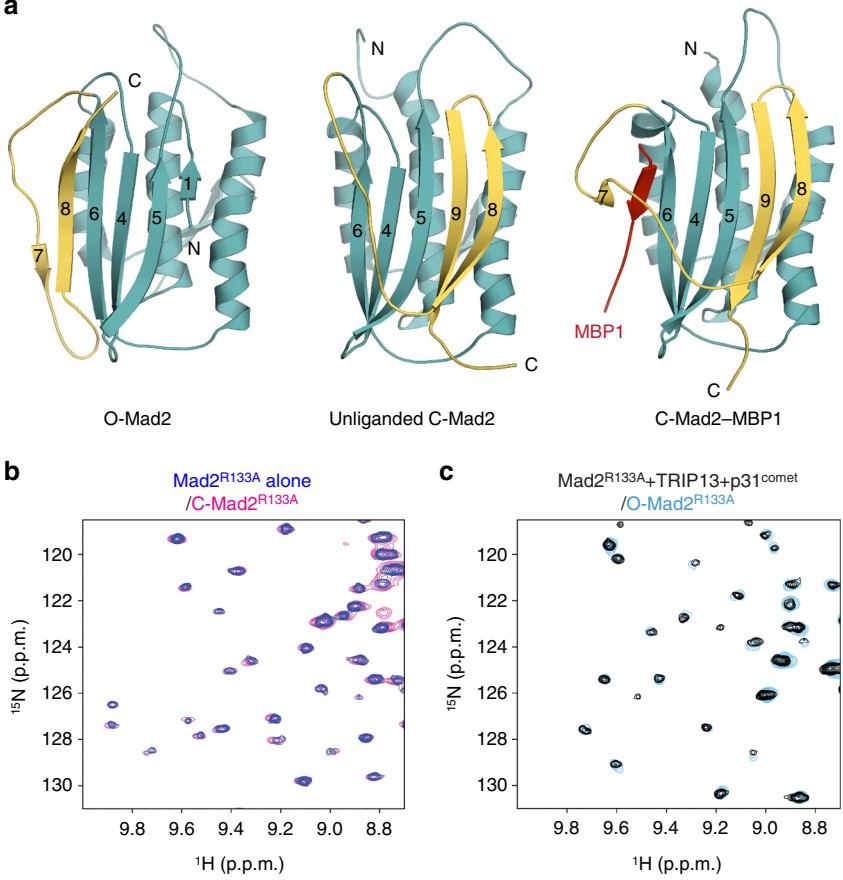

**a**

O-Mad2 Unliganded C-Mad2 C-Mad2–MBP1

**b** Mad2$^{R133A}$ alone /C-Mad2$^{R133A}$

**c** Mad2$^{R133A}$+TRIP13+p31$^{comet}$ /O-Mad2$^{R133A}$

Fig. 1 TRIP13 catalyzes C-Mad2 to O-Mad2 conversion in the presence of p31$^{comet}$. **a** Ribbon diagrams of O-Mad2 (left), unliganded C-Mad2 (middle), and C-Mad2 bound to its high-affinity artificial ligand MBP1 (right). The C-terminal region that undergoes a large conformational change and forms the safety belt structure in C-Mad2 is colored yellow while the rest of the protein is in cyan. Strands in the major β sheet are labeled with their numbers. The N- and C-termini are indicated. All protein structure figures in this study were generated with PyMOL (Schrödinger, LLC; http://www.pymol.org). **b** Overlay of regions of the $^1$H–$^{15}$N HSQC spectra of $^{15}$N-C-Mad2$^{R133A}$ (in magenta) and $^{15}$N-Mad2$^{R133A}$ before the addition of TRIP13 and ΔN35-p31$^{comet}$ (in blue). **c** Overlay of regions of the $^1$H–$^{15}$N HSQC spectra of $^{15}$N-O-Mad2$^{R133A}$ (in cyan) and $^{15}$N-Mad2$^{R133A}$ after the addition of ATP and sub-stoichiometric amounts of TRIP13 and ΔN35-p31$^{comet}$ (in black)

to produce MCC[38–40]. Once formed, the MCC diffuses away from kinetochores and inhibits APC/C$^{Cdc20}$ (refs. [29, 41]).

Kinetochore attachment by microtubules inhibits binding of Mps1 to Ndc80C[18, 19]. This and other mechanisms reduce the levels and activities of Mps1 and downstream checkpoint components at the kinetochore. The C-Mad2-binding protein p31$^{comet}$ binds to the Mad1–C-Mad2 core and limits Mad2 conformational activation, at kinetochores when the checkpoint is on and in the cytosol after Mad1–C-Mad2 is removed from kinetochores during checkpoint inactivation[42–45]. These mechanisms collectively inhibit MCC formation. The existing MCC is then disassembled through APC/C-dependent ubiquitination of Cdc20 and through p31$^{comet}$-mediated MCC disassembly, leading to APC/C$^{Cdc20}$ activation and anaphase onset[46–52]. Recently, the AAA+ ATPase TRIP13 has been shown to collaborate with p31$^{comet}$ to promote MCC disassembly and checkpoint inactivation[53–57]. p31$^{comet}$ acts as an adaptor to recruit C-Mad2 to TRIP13. TRIP13 then induces the conformational change of C-Mad2 to O-Mad2 in a process that requires ATP hydrolysis[55]. The structures of both human TRIP13 and its *Caenorhabditis elegans* homolog PCH-2 have been determined[55, 58]. Chemical crosslinking and functional studies further suggest that TRIP13 might promote the structural transition of Mad2 through engaging and unfolding the N-terminal region of Mad2[58].

In this study, we confirm that, in the presence of p31$^{comet}$, TRIP13 catalyzes the conversion of C-Mad2 to O-Mad2 using nuclear magnetic resonance (NMR) spectroscopy. Proton/deuterium (H/D) exchange experiments further show that TRIP13-catalyzed structural transition of Mad2 does not involve the global unfolding of Mad2, but entails local unfolding of the C-terminal region. We have determined the crystal structure of human TRIP13 in its ADP-bound state, and identified TRIP13 residues critical for p31$^{comet}$–Mad2 binding and for coupling ATP hydrolysis to Mad2 unfolding. Finally, we show that TRIP13–p31$^{comet}$ prevents inhibition of APC/C by MCC components in vitro, but does not restore APC/C activity after APC/C has already been inhibited by MCC. Our studies provide insight into the mechanism of TRIP13-catalyzed conformational change of Mad2 and the molecular processes leading to spindle checkpoint silencing.

## Results

**TRIP13 converts C-Mad2 to O-Mad2 without global unfolding**. To confirm that TRIP13 catalyzed the structural transition of Mad2, we performed NMR analysis of $^{15}$N-labeled Mad2 in the presence or absence of TRIP13 and p31$^{comet}$. For this purpose, we used the monomeric Mad2$^{R133A}$ mutant that retained strong binding to p31$^{comet}$ ($K_d$ 240 nM) and the ability to undergo the O–C structural transition[13, 43]. The $^{1}$H–$^{15}$N heteronuclear single quantum coherence (HSQC) spectrum of Mad2$^{R133A}$ showed that it had adopted the C-Mad2 conformation (Fig. 1b, Supplementary Fig. 1a). We then added ATP and sub-stoichiometric amounts of TRIP13 (1:325 molar ratio of TRIP13 hexamers to Mad2 monomers) and ΔN35-p31$^{comet}$ (1:33 molar ratio of p31$^{comet}$ to Mad2) to the C-Mad2$^{R133A}$ sample, and acquired an HSQC spectrum after 5 h of incubation. The HSQC spectrum of the TRIP13-treated sample was different from that of the starting C-Mad2$^{R133A}$ sample, and matched that of O-Mad2$^{R133A}$ (Fig. 2c, Supplementary Fig. 1b). Therefore, TRIP13–p31$^{comet}$ catalyzes the conversion of unliganded C-Mad2 to O-Mad2 in the presence of ATP.

Because C-Mad2 and O-Mad2 share a common folded core (Fig. 1a), TRIP13–p31$^{comet}$ catalyzed C-Mad2–O-Mad2 conversion does not necessarily require the global unfolding of Mad2. Local unfolding of the C-terminal region of C-Mad2 can

complete this conversion and, in the case of liganded C-Mad2, release the bound ligand. To test whether TRIP13-catalyzed structural transition of Mad2 involved global or local unfolding of Mad2, we performed H/D exchange NMR experiments. Amide protons that form hydrogen bonds in folded structural elements exchange slowly with solvent, and their NMR signals remain visible for hours after the protein is lyophilized and dissolved into D$_2$O. In contrast, amide protons in loop regions or transiently folded regions that do not form stable hydrogen bonds exchange rapidly with solvent, and their NMR signals disappear in D$_2$O.

The HSQC spectrum of C-Mad2$^{R133A}$ in a D$_2$O solution for 2.5 h showed a set of slow-exchanging peaks that mostly belonged to residues in the stably folded core and strand β9 (Fig. 2a, b). A set of slow-exchanging peaks remained at 6 h after the addition of TRIP13, p31$^{comet}$, and ATP (Fig. 2c). The positions of these peaks matched those of peaks in O-Mad2$^{R133A}$, indicating that C-Mad2 had been converted to O-Mad2 during the course of experiment. We mapped these slowing-exchanging amides onto the structures of both C-Mad2 and O-Mad2. The slow-exchanging amides in the presence of TRIP13 and p31$^{comet}$ belonged exclusively to residues in the folded core of Mad2 (Fig. 2d, e). The slow-exchanging amides on β9 and alternating amides on β5 that participated in hydrogen bonding with β9 were no longer protected, indicating that the C-terminal region of C-Mad2 (β8 and β9) were expectedly unfolded during the structural transition. On the other hand, many amide protons in the Mad2 core (β4-6 and αA-C) were protected from exchange with solvent during the TRIP13-catalyzed structural transition from C-Mad2 to O-Mad2. Our results therefore demonstrate that Mad2 does not undergo global unfolding during TRIP13-catalyzed structural conversion.

Because the C-terminal region has to unfold and refold to allow C-Mad2 to convert to O-Mad2, the C-Mad2–O-Mad2 transition catalyzed by TRIP13–p31$^{comet}$ minimally involves the local unfolding of the C-terminal safety belt of C-Mad2. Because not all structural elements of C-Mad2 contain slow-exchanging residues (Fig. 2b), we cannot exclude the possibility of local unfolding of other structural elements, aside from the C-terminal region.

**Structure of human TRIP13 in ADP-bound state**. The structure of PCH-2, the *C. elegans* homolog of TRIP13, has previously been determined, revealing an asymmetric hexameric architecture[55] (Fig. 3a). When viewed from the side, the PCH-2 hexamer forms a slightly curved structure with a convex face and a concave face (Fig. 3a), similar to ClpX[59]. The convex face in ClpX has been shown to bind to substrates, and has generally been referred to as the top face[59]. We will adopt this convention and refer to the convex surface of PCH-2/TRIP13 as the top face.

To understand how TRIP13 recognized p31$^{comet}$ and Mad2, we attempted to determine the structure of the human TRIP13–p31$^{comet}$–Mad2 complex. We subjected human TRIP13 E253A mutant (which was deficient in ATP hydrolysis) bound to ATP and the p31$^{comet}$–Mad2 complex to crystallization, and obtained diffracting crystals. Structure determination, however, revealed that the crystals contained TRIP13 alone in the ADP-bound state (Fig. 3b, Supplementary Fig. 2a). Even though the ATPase-deficient mutant of TRIP13 was used in our studies, the residual ATPase activity of TRIP13 E253A presumably converted ATP to ADP, and contributed to the dissociation of the complex.

Structure of human TRIP13 bound to ADP is highly similar to that of the *C. elegans* PCH-2 promoter bound to ADP, with an RMSD of 1.57 Å for all atoms (Fig. 3b, c). Both contain an N-terminal domain (NTD), a large AAA+ subdomain, and a small AAA+ subdomain. ADP is bound at the interface between the large and small AAA+ subdomains. The structures of the large

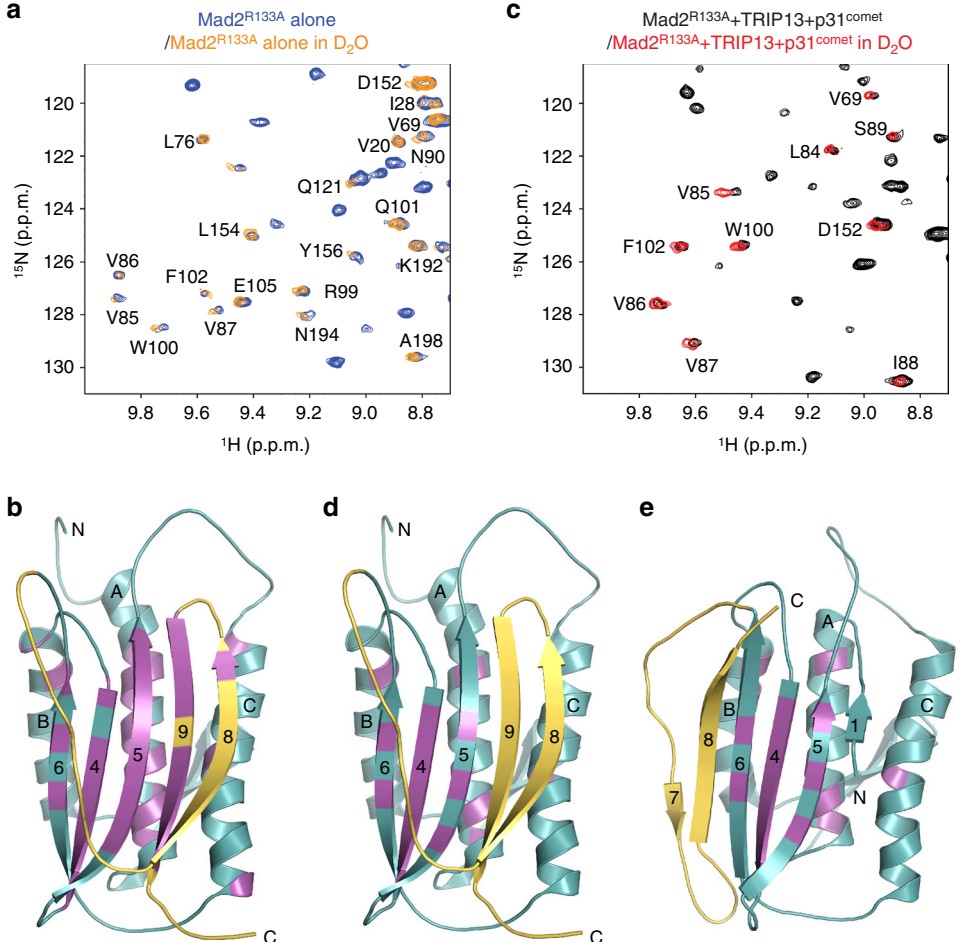

**Fig. 2** TRIP13-catalyzed structural transition of Mad2 does not involve Mad2 global unfolding. **a** Overlay of regions of the $^1H$–$^{15}N$ HSQC spectra of $^{15}N$-labeled C-Mad2$^{R133A}$ in $H_2O$ (blue) and C-Mad2$^{R133A}$ in $D_2O$ (orange). Residues with slow-exchanging amides are labeled. **b** Ribbon diagram of unliganded C-Mad2, with slow-exchanging residues in the absence of TRIP13 colored magenta. Strands in the major β sheet are labeled with their numbers. The N- and C-termini are indicated. **c** Overlay of regions of the $^1H$–$^{15}N$ HSQC spectra of $^{15}N$-labeled Mad2$^{R133A}$ in $H_2O$ (black) and $D_2O$ (red) after the addition of ATP and sub-stoichiometric amounts of TRIP13 and ΔN35-p31$^{comet}$. Residues with slow-exchanging amides are labeled. **d** Ribbon diagram of unliganded C-Mad2, with slow-exchanging residues in the presence of TRIP13 colored magenta. **e** Ribbon diagram of O-Mad2, with slow-exchanging residues in the presence of TRIP13 colored magenta

and small AAA+ subdomains and their relative orientation are virtually identical between human TRIP13 and the ADP-bound protomer of *C. elegans* PCH-2. The only discernable difference is located at the so-called pore loop. This region forms a helix in PCH-2, but does not have visible electron density and is likely flexible in human TRIP13 (Fig. 3c). The pore loop has been implicated in substrate binding and unfolding in other AAA+ ATPases[59, 60]. The structural differences between human and *C. elegans* TRIP13 are consistent with the intrinsically dynamic nature of this loop.

Our structure of human TRIP13 bound to ADP is virtually identical to those of human apo-TRIP13 or TRIP13 bound to ATP[58] (Supplementary Fig. 2b). Because the human TRIP13-structure does not form a closed hexamer (Supplementary Fig. 2c) and has a disordered pore loop, we used the structure of the *C. elegans* PCH-2 hexamer to guide our subsequent mutagenesis and biochemical studies.

**TRIP13 residues critical for p31$^{comet}$–Mad2 stimulation.** To gain insight into the mechanism of TRIP13, we systematically mutated conserved residues in TRIP13 that were surface exposed or partially exposed, and studied the effects of these mutations on

the ATPase activity, Mad2 unfolding, and binding to p31$^{comet}$–Mad2. We describe our findings on the following mutants: W221A, F222A, K227A, K231E, N278A, and G320A. We developed a high-throughput TRIP13 ATPase assay. ATP hydrolysis by TRIP13 wild type (WT) was dose-dependent, and the TRIP13 E253A mutant was deficient in ATP hydrolysis (Supplementary Fig. 3a, b). Consistent with a previous report[55], the ATPase activity of TRIP13 was stimulated by the p31$^{comet}$–C-Mad2 complex, but not by either protein alone (Supplementary Fig. 3c). Similar results were obtained when Mad2 was locked into the C-Mad2 conformation with the L13A mutation, with binding of the artificial peptide ligand MBP1, or with binding of the Cdc20 N-terminal 170-residue fragment (Cdc20N) (Supplementary Fig. 3d). Mad2$^{L13A}$ stimulated TRIP13 more effectively, presumably because it could not stably adopt the O-Mad2 con-former and was expected to go through futile binding/unfolding cycles catalyzed by TRIP13. The ATPase activity of TRIP13 was greatly stimulated at 1:1 molar ratio of TRIP13 hexamer to p31$^{comet}$–Mad2$^{L13A}$ (Supplementary Fig. 3e), suggesting that one TRIP13 hexamer likely engages one p31$^{comet}$–Mad2 heterodimer at a given time.

As reported previously for mouse TRIP13[55], human TRIP13 W221A had much higher basal ATPase activity, nearly equal to

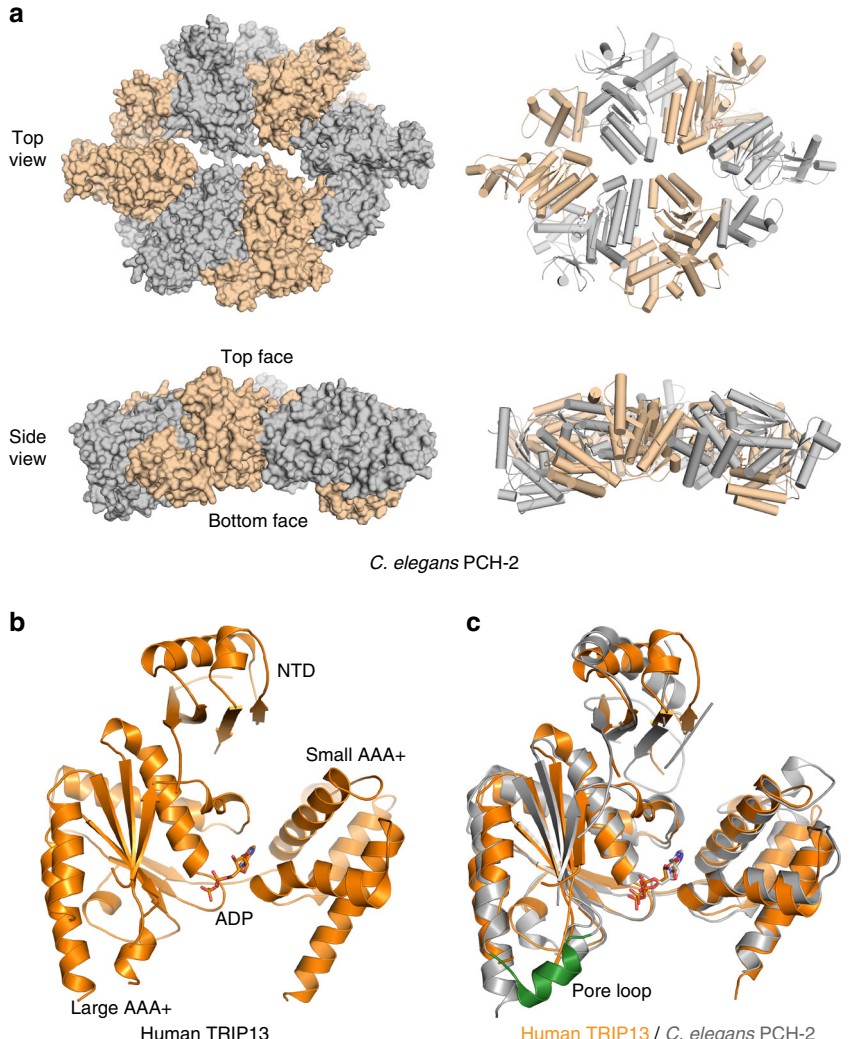

**Fig. 3** Crystal structure of human TRIP13 in its ADP-bound state. **a** Surface and ribbon diagrams of *C. elegans* PCH-2 (PDB ID: 4XGU) in top and side views. Neighboring protomers are shown in different colors. **b** Ribbon diagram of the crystal structure of human TRIP13, with the bound ADP shown in sticks. **c** Overlay of the ribbon diagrams of the structures of human TRIP13 (orange) and the ADP-bound protomer of *C. elegans* PCH-2 (gray; except the pore loop, which is colored green), with the bound ADP molecules shown in sticks

the wild-type TRIP13 activity stimulated by p31[comet]–Mad2 (Fig. 4a). The ATPase activity of TRIP13 W221A was not further stimulated by p31[comet]–Mad2. TRIP13 W221 is located in the pore loop (Fig. 4b), a structural element of AAA+ ATPases thought to be responsible for substrate translocation into the pore of the hexamer[59]. In contrast to W221A, mutation of F222, a neighboring residue in the pore loop, did not reduce the basal ATPase activity of TRIP13, but abolished the stimulation by p31[comet]–Mad2 (Fig. 4a). Thus, the pore loop of TRIP13 serves two function: (1) restraining the basal ATPase activity of TRIP13 and (2) coupling ATP hydrolysis to p31[comet]–Mad2 binding or the conformational change of Mad2. F222 is specifically involved in the second function.

The α2 helix in the large AAA+ subdomain is located on the top face of TRIP13 and is adjacent to the pore loop[55, 61] (Fig. 4b). Mutations of K227 and K231, two residues in the α2 helix, both reduced the stimulated activity of TRIP13 by p31[comet]–Mad2, despite having opposite effects on the basal ATPase activity of TRIP13 (Fig. 4a). Intriguingly, mutation of N278, a residue in the α3 helix in the large AAA+ subdomain[61] (Fig. 4b), also abolished the stimulation of TRIP13 by p31[comet]–Mad2 (Fig. 4b). Finally, mutation of G320 in the hinge between the large and small AAA+

subdomains greatly reduced the ATPase activity of TRIP13 and its stimulation by p31[comet]–Mad2 (Fig. 4a, b). Thus, we have identified a set of residues in diverse structural elements of TRIP13 that is critical for coupling ATP hydrolysis to substrate binding or unfolding.

We developed a beads-based assay to monitor the disassembly of C-Mad2–ligand complexes by TRIP13 (Fig. 5a). TRIP13 WT was capable of efficiently dissociating Mad2 from MBP1 in the presence of ATP and p31[comet] (Fig. 5b) and in a dose-dependent manner (Supplementary Fig. 4a). The ATPase-deficient mutant TRIP13 E253A was unable to dissociate Mad2, but bound strongly to p31[comet]–Mad2–MBP1 in the presence of ATP. Interestingly, TRIP13 at 25 nM exhibited considerable ATPase activity (Fig. 4a), but no dissociation of Mad2 was observed at this concentration (Supplementary Fig. 4a). For unknown reasons, only at concentrations above 500 nM, TRIP13 was able to efficiently dissociate Mad2. We next tested whether the TRIP13 mutations affected the ability of TRIP13 to release C-Mad2 from bound ligands. TRIP13 W221A, F222A, K227A, K231E, and N278A, which were deficient in p31[comet]–Mad2 stimulation, all failed to dissociate Mad2 from MBP1 (Fig. 5c) or Cdc20N (Supplementary Fig. 4b). Thus, the coupling between ATP

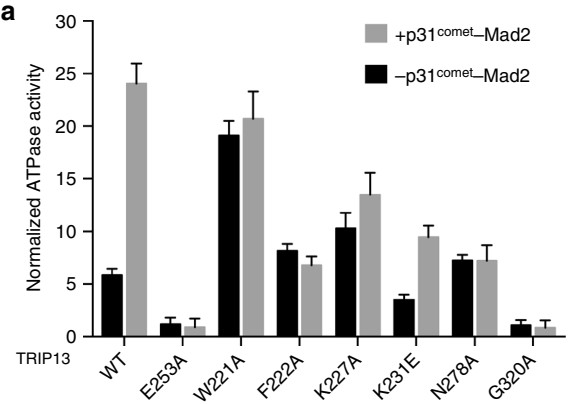

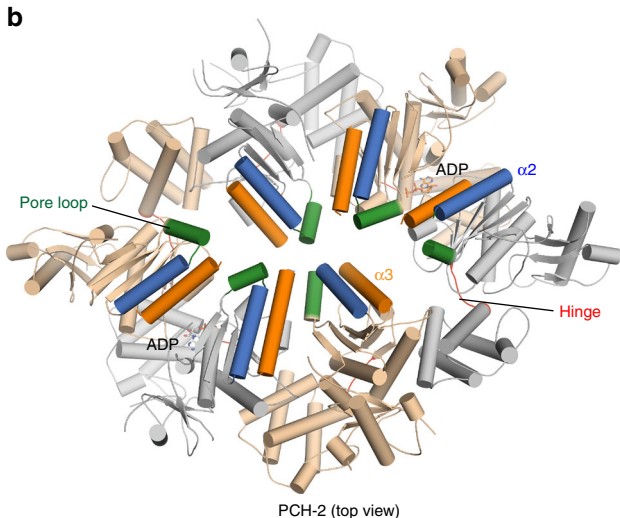

**Fig. 4** Identification of TRIP13 mutants with defective p31$^{comet}$–Mad2 stimulation. **a** Normalized ATPase activities of the indicated TRIP13 proteins at 25 nM with (+) or without (−) 50 nM ΔN35-p31$^{comet}$–Mad2$^{L13A}$. Mean ± SD; $n = 9$. **b** Cartoon diagram of *C. elegans* PCH-2 in the top view. The pore loop, the α2 helix, the α3 helix, and the hinge are shown in green, blue, orange, and red, respectively. The bound ADP molecules are shown in sticks

hydrolysis and p31$^{comet}$–Mad2 binding or Mad2 unfolding is critical for releasing Mad2 from ligands.

**The hinge impacts TRIP13 activity and oligomerization.** Surprisingly, TRIP13 G320A, which at 25 nM showed little ATPase activity (Fig. 4a), was able to dissociate Mad2 from MBP1 or Cdc20 (Fig. 5c, Supplementary Fig. 4b). This apparent discrepancy prompted us to perform further experiments on TRIP13 G320A. TRIP13 G320 is located at the hinge between its large and small AAA+ subdomains (Fig. 5d). Because the Mad2 dissociation assay was performed with 500 nM TRIP13, we tested whether TRIP13 G320A had ATPase activity at higher concentrations. Indeed, even though TRIP13 G320A at 25 nM had undetectable ATPase activity in the presence or absence of p31$^{comet}$–Mad2, this mutant had substantial basal activities at 100 nM and 200 nM concentrations (Fig. 5e), which were further stimulated by p31$^{comet}$–Mad2. This likely explains the ability of G320A at high concentrations to dissociate Mad2 from ligands.

One possibility for the lack of ATPase activity of G320A at low concentrations is that the mutation weakens the oligomerization of TRIP13 and prevents it from reaching/maintaining the functional hexameric state. To test this possibility, we compared

the elution profiles of TRIP13 WT and G320A in the presence of ATP on a gel filtration column. Both proteins had very similar fractionation profiles, and predominantly formed oligomers (Supplementary Fig. 5). The fractionation profiles of TRIP13 WT in the presence of ATP were very similar to those reported previously[58]. Thus, in the presence of ATP, the G320A mutation does not appear to disrupt the oligomerization of TRIP13.

We next subjected TRIP13 WT to analytical ultracentrifugation sedimentation velocity studies in the presence of 100 μM ATP. While interpretable concentration profiles could be collected from such samples, the resulting $c(s)$ distributions demonstrated a large degree of polydispersity in $s_{20,w}$ values greater than 5 s. Thus, consistent with the broad fractionation profiles on gel filtration, ATP binding does not induce the formation of discrete TRIP13 oligomers in vitro, possibly due to the nucleotide-driven conformational dynamics of TRIP13. Interestingly, in the absence of ATP, TRIP13 WT at high concentrations displayed sedimentation behavior consistent with an equilibrium between discrete species (Fig. 5f). By contrast, TRIP13 G320A was monomeric under all concentrations tested. Thus, TRIP13 G320A is deficient for oligomerization in the nucleotide-free state, even though the hinge is not located in the hexamer interface.

Previous studies have shown that not all six protomers of AAA + ATPases bind to and hydrolyze nucleotides at the same time[59]. The relative positions of the small and large AAA+ subdomains in a given protomer are different between its nucleotide-bound and nucleotide-free states[55]. Thus, the flexibility of the hinge might enable relative movements between the two subdomains during the ATPase cycle. These movements not only drive substrate remodeling, but also maintain the functional hexamer state of the enzyme. We propose that the G320A mutation interferes with conformational dynamics of TRIP13, weakens TRIP13 oligomerization, and reduces the ATPase activity of TRIP13. Sequence alignment of the hinge regions of TRIP13 and two other eukaryotic AAA+ ATPases, p97/VCP and NSF, revealed a conserved motif with the consensus of φGXP (φ, a hydrophobic residue; X, any residue) (Fig. 5g). Mutations affecting the length of the hinge in ClpX uncouple the ATPase activity from substrate unfolding[62]. We speculate that hinge flexibility in AAA+ ATPases may be generally important for the conformational dynamics between the large and small AAA+ subdomains during the ATPase cycle.

**Identification of p31$^{comet}$–Mad2-binding residues on TRIP13.** The TRIP13 mutants deficient for stimulation by p31$^{comet}$–Mad2 and for Mad2 release from ligands might have lost their ability to bind p31$^{comet}$–Mad2 or be defective in coupling ATP hydrolysis to Mad2 unfolding. To distinguish these possibilities, we wished to directly assay the binding of p31$^{comet}$–Mad2 to TRIP13. We observed that TRIP13 E253A, which was deficient in ATP hydrolysis and Mad2 dissociation, was able to bind to p31$^{comet}$–Mad2–MBP1 beads in the Mad2 dissociation assay (Fig. 5b, c). We hypothesized that the ATP-bound state of TRIP13 might bind more strongly to p31$^{comet}$–Mad2. Indeed, addition of ATPγS, a slowly hydrolyzable ATP analog, did not support Mad2 dissociation, but greatly enhanced the binding of TRIP13 WT to beads bound with p31$^{comet}$–Mad2–MBP1 (Supplementary Fig. 6a). Addition of ADP did not enhance TRIP13 WT binding to p31$^{comet}$–Mad2–MBP1, and reduced the binding of TRIP13 E253A, presumably because ADP displaced some ATP bound to this mutant. These results show that p31$^{comet}$–Mad2 prefers to bind to the ATP-bound state of TRIP13. The use of ATPγS in place of ATP converts the Mad2 dissociation assay into an assay that can monitor the binding between TRIP13 and p31$^{comet}$–Mad2.

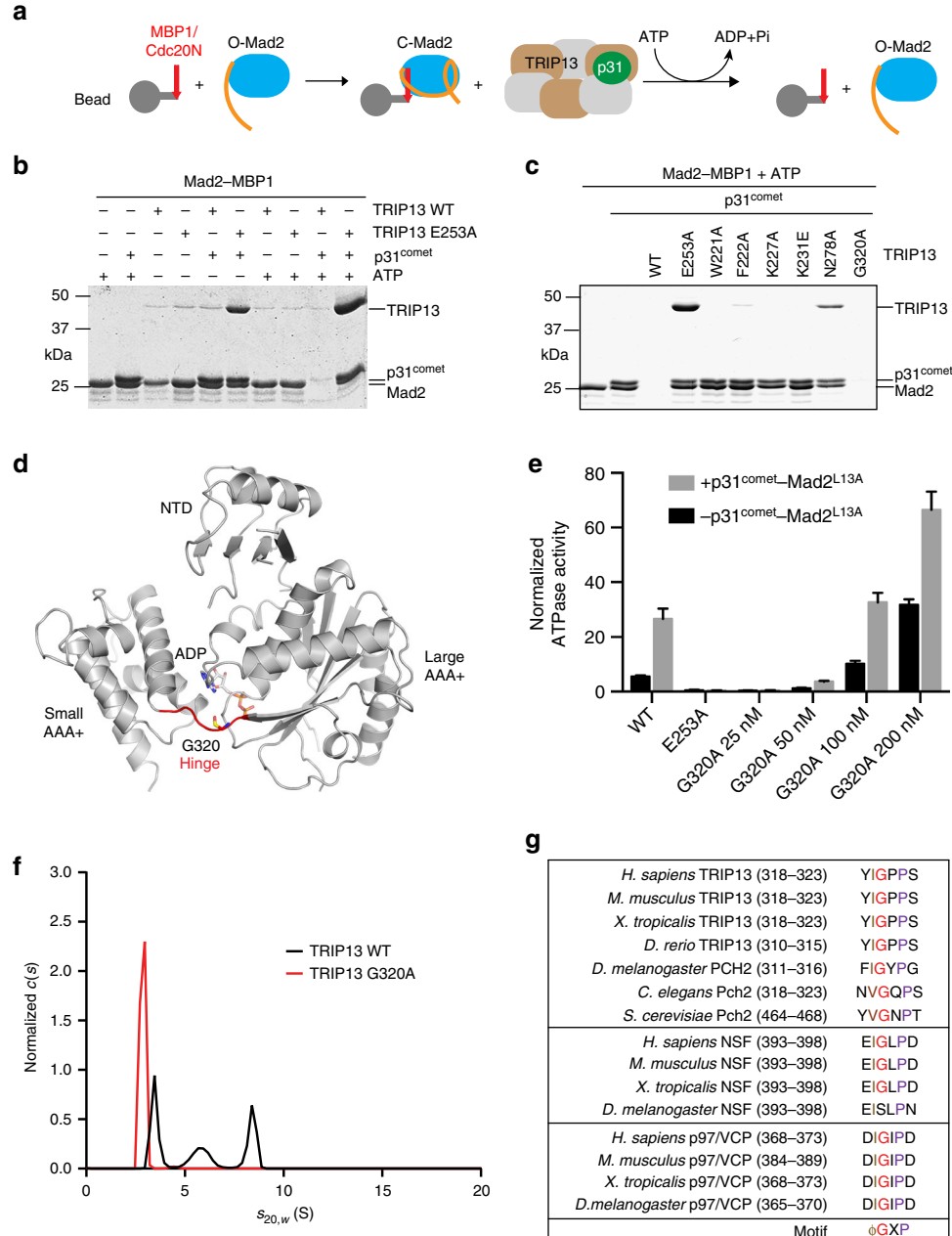

**Fig. 5** The TRIP13 hinge is required for conformational dynamics and oligomerization. **a** Schematic drawing of the Mad2 dissociation assay. **b** MBP1-coupled beads were first incubated with Mad2 and then incubated with or without TRIP13 wild type (WT) or E253A (1 μM), ΔN35-p31$^{comet}$ (2 μM), or ATP (1 mM). Proteins bound to beads were analyzed by SDS-PAGE and stained with Coomassie blue. **c** MBP1-coupled beads were first incubated with Mad2 and then incubated with TRIP13 wild type (WT) and the indicated mutants (500 nM) in the presence of ΔN35-p31$^{comet}$ (1 μM) and ATP (1 mM). Proteins bound to beads were analyzed by SDS-PAGE and stained with Coomassie blue. **d** Ribbon diagram of human TRIP13, with the hinge connecting the large and small AAA+ subdomains colored red. ADP and G320 are shown in sticks. **e** ATPase activities of TRIP13 WT at 25 nM and G320A at the indicated concentrations with (+) or without (−) ΔN35-p31$^{comet}$-Mad2$^{L13A}$. The concentrations of p31$^{comet}$-Mad2 were kept at 2:1 molar ratio of those of TRIP13. Mean ± SD; $n = 8$. **f** Sedimentation velocity analysis of TRIP13 WT and G320A in the absence of ATP. **g** Sequence alignment of the hinge region of TRIP13, the D1 domain NSF, and the D1 domain p97/VCP from different organisms

We next tested the binding of the TRIP13 mutants to p31$^{comet}$–Mad2 with this assay. TRIP13 W221A, F222A, K227A, and K231E were deficient in binding to p31$^{comet}$–Mad2–MBP1 (Fig. 6a) or p31$^{comet}$–Mad2–Cdc20N (Supplementary Fig. 6b), suggesting that these residues at the pore loop and the α2 helix mediate binding of p31$^{comet}$–Mad2 to TRIP13 (Fig. 6b). Therefore, p31$^{comet}$–Mad2 likely binds to the top face of TRIP13 through charged interactions with the α2 helix and through interactions with the pore loop.

Recently, Corbett and workers suggested that TRIP13 engages the N-terminal region of Mad2 through the pore loop and unfolds Mad2 from the N-terminus[58]. The most crucial evidence for Mad2 N-terminal unfolding by TRIP13 is the crosslinking data performed with the TRIP13 W221C mutant. This cross-linking appeared to show that the N-terminal region of Mad2 makes direct contact with the TRIP13 pore loop. On the other hand, we have shown that the W221A mutation abrogates binding of TRIP13 to p31$^{comet}$–Mad2 in the presence of ATPγS

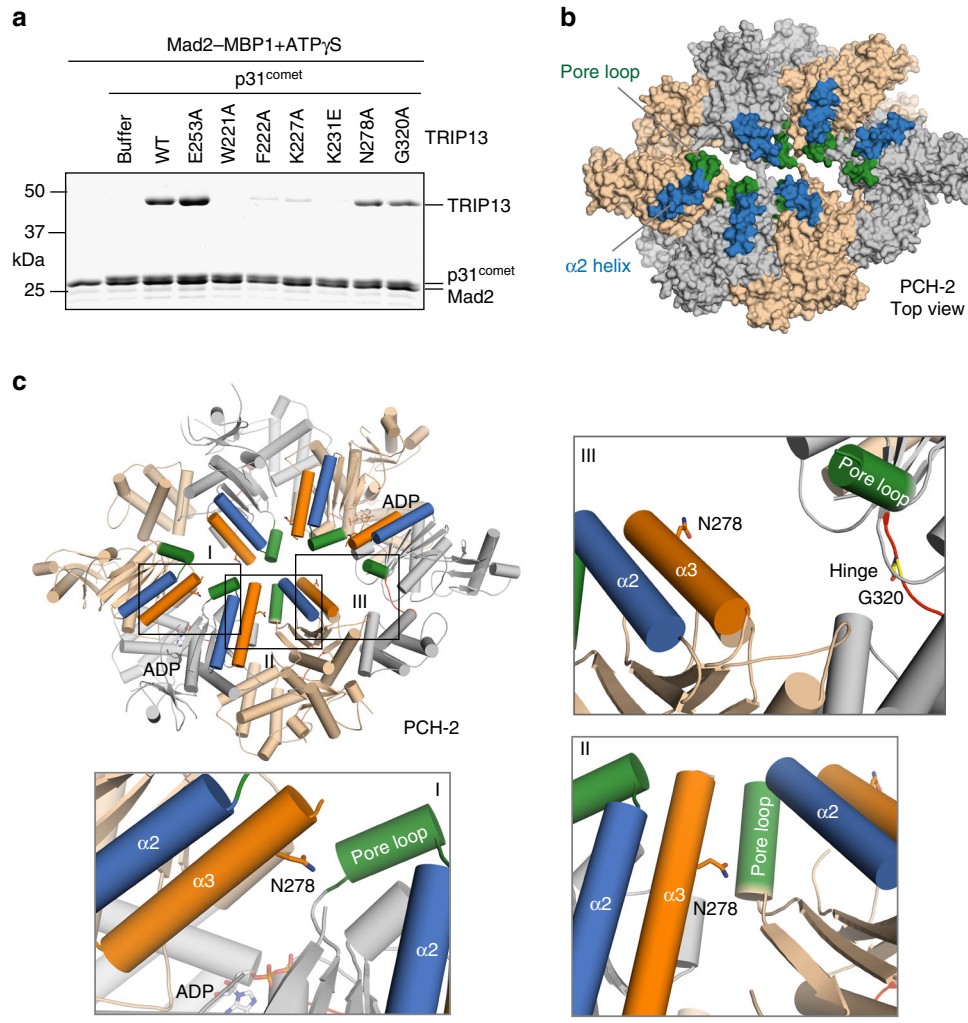

**Fig. 6** Identification of a p31$^{comet}$–Mad2-binding site on TRIP13. **a** MBP1-coupled beads were first incubated with Mad2, and then incubated with TRIP13 wild type (WT) or the indicated mutants (500 nM), ΔN35-p31$^{comet}$ (1 μM), and ATPγS (1 mM). Proteins bound to beads were analyzed by SDS-PAGE and stained with Coomassie blue. **b** Surface diagram of *C. elegans* PCH-2 in the top view, with the pore loop and α2 helix colored green and blue, respectively. **c** Cartoon diagram of *C. elegans* PCH-2, with zoomed-in views of the boxed interfaces (I–III) shown. N278, ADP, and G320 are shown in sticks. The pore loop, the α2 helix, and the α3 helix are colored green, blue, and orange, respectively. Each of the three types of interfaces is present twice in the PCH-2 hexamer

(Fig. 6a, Supplementary Fig. 6b). We wondered whether the W221C mutant used in the Corbett study was similarly defective in binding to p31$^{comet}$–Mad2. We thus made the TRIP13 W221C mutant (Supplementary Fig. 6c). Expectedly, the W221C mutant behaved very similarly to the W221A mutant. It had higher basal ATPase activity, which was not further stimulated by p31$^{comet}$–Mad2 (Supplementary Fig. 6d). W221C also could not dissociate Mad2 from its ligand MBP1 (Supplementary Fig. 6e). More importantly, W221C was defective in binding to p31$^{comet}$–Mad2 (Supplementary Fig. 6f). Therefore, the installation of a crosslinkable residue in the pore loop destroys the function of this loop. Although the crosslinking experiments were performed with sub-stoichiometric amount of W221C mixed with TRIP13 WT, it remains possible that the mutated pore loop does not form functional contact with Mad2 even when it is present as a single copy in the hexamer.

TRIP13 G320A expectedly retained binding to p31$^{comet}$–Mad2 (Fig. 6a, Supplementary Fig. 6b), as it was capable of dissociating Mad2 from ligands in the presence of ATP. The weakened binding of TRIP13 G320A to p31$^{comet}$–Mad2 is likely an indirect consequence of altered conformational dynamics of the mutant.

Surprisingly, the TRIP13 N278A mutant, which was deficient in promoting Mad2 dissociation from ligands and whose ATPase activity was not stimulated by p31$^{comet}$–Mad2, retained substantial binding to p31$^{comet}$–Mad2. Thus, the N278 mutation uncouples p31$^{comet}$–Mad2 binding from ATP hydrolysis and subsequent conformational change of Mad2. N278 is therefore critical for harnessing the energy of ATP hydrolysis to switch the conformation of Mad2. TRIP13 N278 is equivalent to PCH-2 N278, which is located at the interface between adjacent PCH-2 protomers. It is in close proximity to the pore loop at four such interfaces between one nucleotide-bound protomer and one nucleotide-free protomer (I and II), but is far away from the pore loop at the other two interfaces between two nucleotide-free protomers (III) (Fig. 6c). We speculate that N278 engages with the pore loop in specific nucleotide states of PCH-2/TRIP13 when it is bound to p31$^{comet}$–Mad2, and facilitates the dynamic movements of the pore loop during ATP hydrolysis.

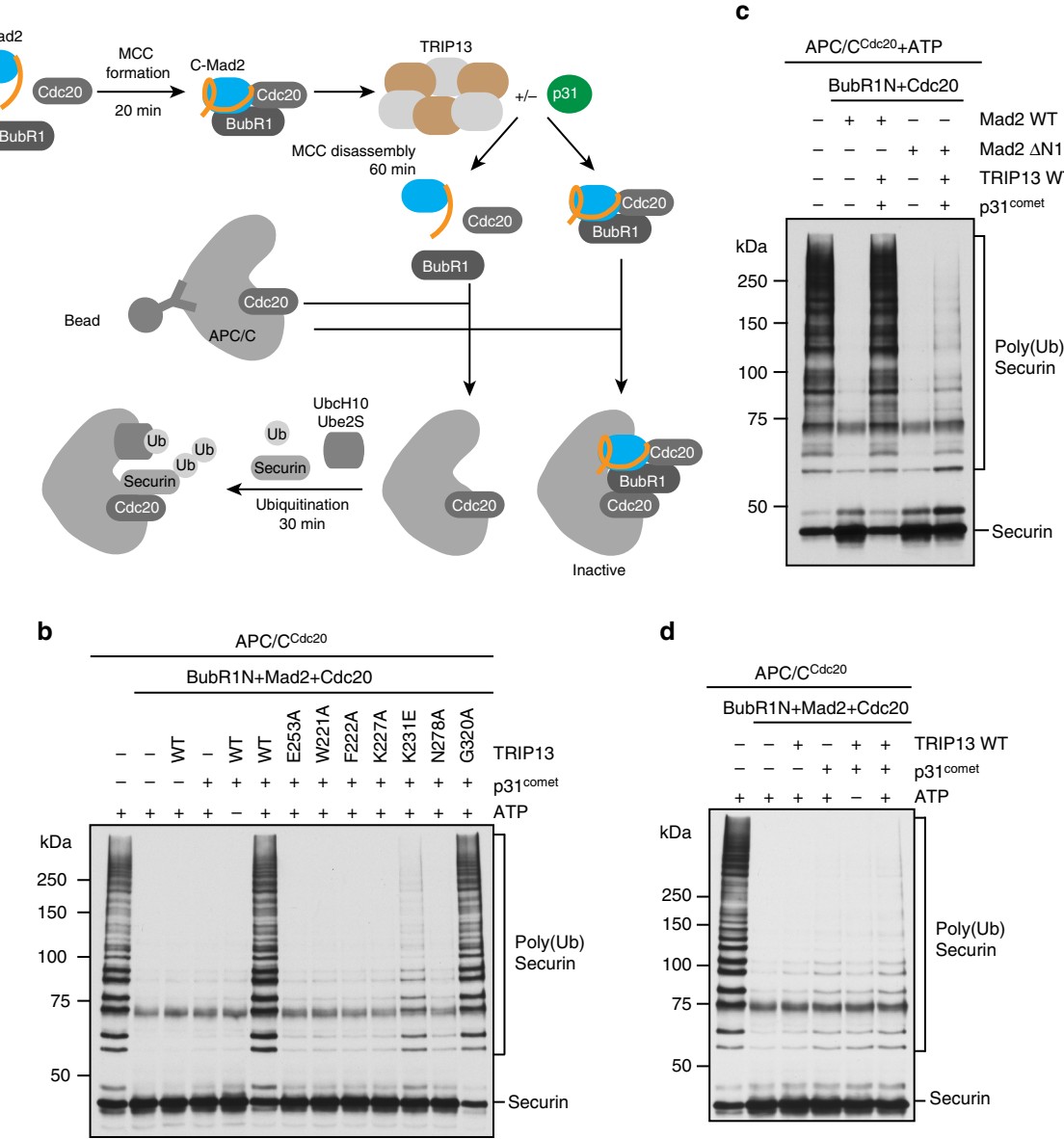

**Fig. 7** TRIP13 prevents APC/C$^{Cdc20}$ inhibition by MCC components. **a** Schematic diagram of the APC/C ubiquitination assay in the absence or presence of MCC components, TRIP13, or p31$^{comet}$. **b** Ubiquitination of securin-Myc by APC/C$^{Cdc20}$ in the presence of MCC components (780 nM BubR1N, 1 μM Mad2, and 600 nM Cdc20), TRIP13 wild type (WT) or the indicated mutants (50 nM), and ΔN35-p31$^{comet}$ (1 μM). MCC components were pre-incubated with TRIP13, p31$^{comet}$, and ATP (1 mM). The protein mixture was then added to APC/C$^{Cdc20}$. The ubiquitination reaction mixtures were blotted with the anti-Myc antibody. **c** Same as in **b** except that Mad2 ΔN10 and only TRIP13 WT were used. **d** Ubiquitination of securin-Myc by APC/C$^{Cdc20}$ that was first incubated with MCC components (780 nM BubR1N, 1 μM Mad2, and 600 nM Cdc20) for 20 min and then incubated with TRIP13 WT (50 nM) and ΔN35-p31$^{comet}$ (1 μM). The ubiquitination reaction mixtures were blotted with the anti-Myc antibody

**Reconstitution of TRIP13-dependent MCC inactivation in vitro**. MCC disassembly has been proposed to be the key function of TRIP13 in checkpoint silencing[53, 54, 56]. We tested whether TRIP13, with the help of p31$^{comet}$, could inactivate MCC in vitro. As shown previously[28, 29, 63], incubation of recombinant BubR1N (residues 1–370), dimeric Mad2 WT, and full-length Cdc20 produced a minimal, functional MCC, which inhibited the ubiquitination of securin by APC/C$^{Cdc20}$ (Fig. 7a, b). Addition of TRIP13, p31$^{comet}$, and ATP to the mixture of MCC components (prior to the addition of APC/C$^{Cdc20}$) completely restored the ubiquitination activity of APC/C$^{Cdc20}$ (Fig. 7b), indicating that TRIP13 was able to inactivate MCC. As expected, TRIP13-dependent MCC inactivation required p31$^{comet}$ and ATP. The ATPase-deficient TRIP13 E253A was unable to restore APC/

C$^{Cdc20}$ ubiquitination activity. The TRIP13 mutants defective for Mad2 dissociation from peptide ligands were also unable to inactivate MCC. TRIP13 G320A, which was capable of releasing Mad2 from ligands, was able to inactivate MCC to allow APC/C$^{Cdc20}$-mediated ubiquitination of securin. The perfect correlation between the abilities of TRIP13 mutants to release Mad2 from ligands and to inactivate MCC strongly suggest that TRIP13 inactivates MCC through promoting the C-Mad2 to O-Mad2 conversion and disrupting the C-Mad2–Cdc20 interaction.

We next tested whether the N-terminal region of Mad2 was required for TRIP13-dependent inactivation of MCC. Mad2 ΔN10 could be incorporated into a functional MCC that inhibited APC/C$^{Cdc20}$ (Fig. 7c). In contrast to MCC containing Mad2 WT, MCC containing Mad2 ΔN10 was not efficiently

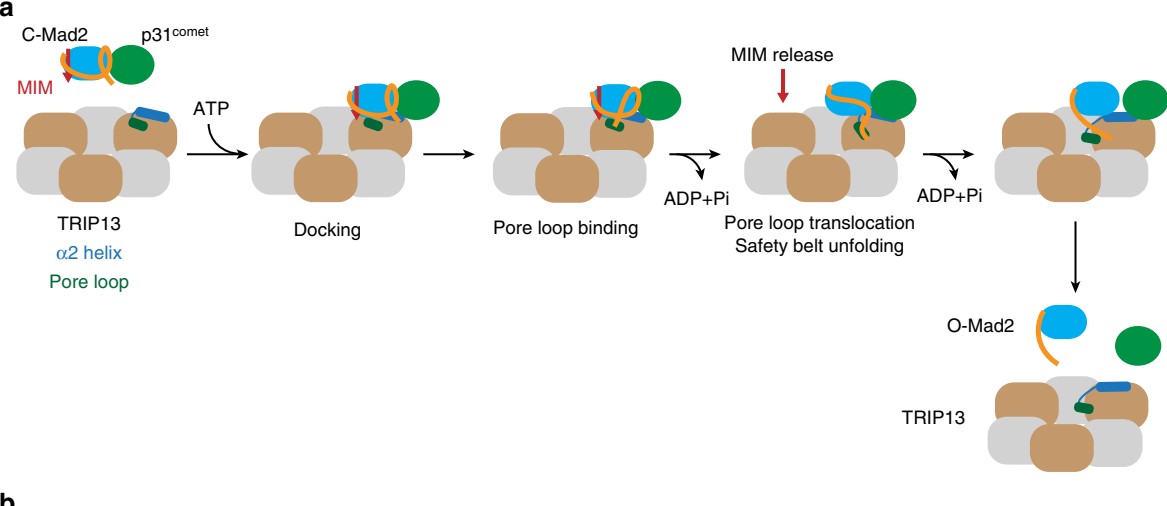

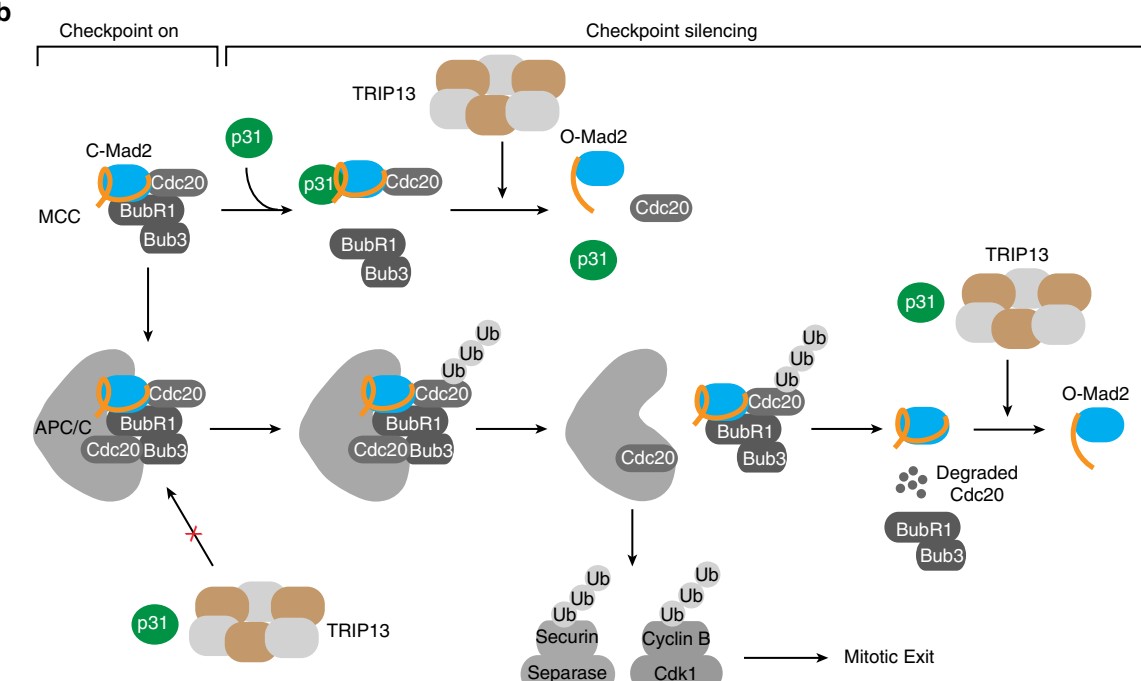

**Fig. 8** Mechanisms of TRIP13-dependent Mad2 conformational change and checkpoint silencing. **a** Model of TRIP13-catalyzed conversion of C-Mad2 to O-Mad2. The p31[comet]–C-Mad2–MIM complex docks on the α2 helix of TRIP13. The pore loop of TRIP13 engages the C-terminus of Mad2. ATP hydrolysis drives the translocation of the pore loop and the associated Mad2 C-terminus, leading to the local unfolding of the C-terminal safety belt of C-Mad2. This Mad2 local unfolding releases MIM and disrupts the p31[comet]–Mad2 interaction. Mad2 refolding produces O-Mad2. Both p31[comet] and O-Mad2 dissociate from TRIP13. **b** Model of TRIP13-dependent checkpoint silencing. Upon spindle checkpoint activation, MCC is produced at unattached kinetochores and diffuses into the cytosol to inhibit APC/C[Cdc20]. During checkpoint silencing, the production of MCC is attenuated. p31[comet] binds to C-Mad2 in MCC and displaces BubR1–Bub3 from MCC. TRIP13 then binds to p31[comet]–C-Mad2–Cdc20, disassembles the C-Mad2–Cdc20 complex, and converts C-Mad2 to O-Mad2. Because TRIP13 cannot act on MCC already bound to APC/C[Cdc20], MCC must first be released from APC/C[Cdc20] through alternative mechanisms. Cdc20 ubiquitination is a potential MCC-releasing mechanism. Proteasome-mediated degradation of ubiquitinated Cdc20 then disassembles MCC. C-Mad2 that persists during this process can be recognized by p31[comet] and TRIP13, and converted to O-Mad2. Collectively, these mechanisms reduce the levels of MCC and promote the activation of APC/C[Cdc20], which ubiquitinates securin and cyclin B1 to trigger chromosome segregation and mitotic exit

inactivated by TRIP13, and APC/C activity was not fully restored. Furthermore, Mad2 ΔN10 retained its binding to p31[comet], but the Mad2 ΔN10–p31[comet] complex was deficient in TRIP13 binding (Supplementary Fig. 6g). Therefore, consistent with the results of Corbett and coworkers[58], the N-terminal region of Mad2 is required for TRIP13 binding and proper MCC disassembly. This result does not, however, indicate that the N-terminus of Mad2 directly engages the pore loop of TRIP13 or

that the unfolding of Mad2 by TRIP13 starts at the N-terminus. We note that our Mad2 ΔN10 protein contained a 12-residue tag in place of the native N-terminal 10 residues of Mad2. It is formally possible that the extra two residues in length in the Mad2 ΔN10 protein, not the change in amino acid sequence, accounts for the weakened binding to TRIP13.

Finally, we tested whether TRIP13–p31[comet] could inactivate MCC that was already bound to APC/C[Cdc20]. We pre-incubated

the MCC components with APC/C[Cdc20] and then added TRIP13, p31[comet], and ATP, before performing the ubiquitination assay. TRIP13–p31[comet] was unable to relieve the inhibition of APC/C[Cdc20] by the pre-bound MCC (Fig. 7c). Therefore, our results indicate that TRIP13 can catalyze the disassembly of free MCC, but cannot effectively inactivate MCC already bound to APC/C[Cdc20].

## Discussion

The asymmetric hexameric ring structure of TRIP13 is reminiscent of that of the bacterial unfoldase ClpX[55, 59, 64]. ClpX binds to substrates with its convex, top surface, and is believed to completely unfold substrates and thread them through an inner pore that is lined by the pore loop conserved in AAA+ ATPases[59]. The unfolded substrates that emerge from the pore are then degraded by the ClpP protease, which is bound to the concave, bottom surface of ClpX[59]. The substrates of ClpX are bacterial proteins, which are not structurally related to Mad2. We have found that, similar to ClpX, TRIP13 binds to p31[comet]–Mad2 on its convex, top surface through charged interactions with the α2 helix in the large AAA+ subdomain and through engaging the pore loop. Because blocking ATP hydrolysis by mutations or ATPγS enhances p31[comet]–Mad2 binding to TRIP13, p31[comet]–Mad2 likely binds to the ATP-bound promoters of TRIP13.

Unlike ClpX, however, TRIP13-catalyzed conformational change does not involve the complete, global unfolding of Mad2, as slowly exchanging amide protons are protected from $D_2O$ during the entire process. We propose that the p31[comet]–Mad2–MIM complex docks on TRIP13 protomers in the ATP-bound state and at a site involving the α2 helix (Fig. 8a). The C-terminus of Mad2 then engages the pore loop of TRIP13 and relieves the autoinhibition of the ATPase activity. ATP hydrolysis drives the movement of the pore loop, which tugs at the Mad2 C-terminus. The translocation of the Mad2 C-terminus is further supported by other structural elements, such as N278 in α3. The local unfolding of the C-terminal safety belt of Mad2 releases the bound MIM and disrupts the p31[comet]–Mad2 interaction. As TRIP13 does not bind strongly to either Mad2 or p31[comet], the breakup of the p31[comet]–Mad2 complex causes both proteins to be released from TRIP13. TRIP13 can then bind another p31[comet]–Mad2–MIM complex to repeat the process.

At present, we do not know whether the local unfolding of Mad2 C-terminal region involves the threading of this region through the central pore of the TRIP13 hexamer. If it does, mechanisms must exist to release the partially unfolded Mad2 from TRIP13 so that the stable core of Mad2 is not threaded through the pore and unfolded. Alternatively, ATP-driven translocation events of the Mad2 C-terminus may suffice to unravel the C-terminal safety belt of Mad2 and occur solely on the top face of TRIP13.

A recent study suggests that the N-terminal region of Mad2 directly engages the pore loop of TRIP13 and that Mad2 unfolding by TRIP13 starts at the N-terminus[58]. The key evidence supporting these claims involves crosslinking experiments with the TRIP13 W221C mutant as a sub-stoichiometric mixture with WT. We have now shown that this pore loop mutant as a hexamer is defective in p31[comet]–Mad2 binding and in Mad2 unfolding. It is possible that the mutated pore loop, even as a single copy in a hexamer, does not make functional contact with Mad2. On the other hand, consistent with the previously reported cellular data[58], the Mad2 N-terminus is required for TRIP13 binding and for TRIP13-catalyzed disassembly of MCC. Although it makes more intuitive sense that TRIP13 initiates Mad2 unfolding locally from the C-terminus, our results do not disprove that TRIP13 unfolds Mad2 locally from the N-terminus.

Our reconstitution of TRIP13–p31[comet]-dependent MCC inactivation in vitro has provided insight into the mechanism by which TRIP13 promotes spindle checkpoint silencing. Perhaps our most important finding is that TRIP13–p31[comet] only inactivates free MCC prior to its binding to APC/C[Cdc20], but not MCC already bound to APC/C[Cdc20]. We propose that TRIP13–p31[comet] disassembles free MCC in a step-wise fashion (Fig. 8b). Because p31[comet] and BubR1 bind to overlapping interfaces on C-Mad2 and compete for C-Mad2 binding[45, 65, 66], p31[comet] binding to C-Mad2 disrupts the integrity of MCC and releases BubR1. The p31[comet]–C-Mad2–Cdc20 complex is then recognized by TRIP13. TRIP13-catalyzed conversion of C-Mad2 to O-Mad2 dissociates Mad2 from Cdc20, causing the complete disassembly of MCC. In this way, TRIP13–p31[comet] reduces the cellular concentration of free MCC and helps to turn off the spindle checkpoint.

Our results suggest that additional mechanisms are required to inactivate MCC bound to APC/C and promote checkpoint silencing. One such mechanism is APC/C[Cdc20]-dependent ubiquitination of the Cdc20 molecule that is part of the MCC[47, 48]. Cdc20 ubiquitination is likely sufficient to release MCC from APC/C, which can then be acted on by TRIP13–p31[comet] (Fig. 8b). Alternatively, degradation of ubiquitinated Cdc20 by the proteasome can cause the complete disassembly of MCC.

In addition to promoting MCC disassembly, TRIP13–p31[comet] is needed to revert the spontaneously converted C-Mad2 back to O-Mad2. Mad2 released by Cdc20 degradation that is still in the C-Mad2 conformer may also be converted to O-Mad2 by TRIP13–p31[comet] (Fig. 8b). TRIP13–p31[comet] may therefore be required to maintain the pool of O-Mad2 in the cell by converting free, unliganded C-Mad2 back to the O-Mad2. Failure to do so causes premature accumulation of C-Mad2 and spindle checkpoint defects[67], as O-Mad2 is the Mad2 conformer that is activated by Mad1–Mad2 at kinetochores to become I-Mad2, which is subsequently incorporated into MCC.

In conclusion, using a combination of biophysical and biochemical approaches, we have demonstrated that TRIP13–p31[comet] catalyzes the conversion of C-Mad2 to O-Mad2 without the global unfolding of Mad2. We have defined the functional roles of multiple structural elements of TRIP13 in coupling ATP hydrolysis to Mad2 unfolding. Finally, we have reconstituted TRIP13–p31[comet]-dependent MCC inactivation in vitro, leading to new testable models of how TRIP13 and p31[comet] promote spindle checkpoint silencing and the metaphase–anaphase transition.

## Methods

**Protein expression and purification.** BL21 (DE3) *Escherichia coli* cells were transformed with plasmids encoding His$_6$-tagged human TRIP13 WT or mutants, cultured in Terrific Broth until the $OD_{600}$ reached 1.0, and induced overnight at 18 °C with 0.2 mM isopropyl β-D-1-thiogalactopyranoside (IPTG). Collected cells were resuspended in the TRIP13 Purification Buffer (50 mM $NaH_2PO_4$·$2H_2O$ pH 8.0, 300 mM NaCl, 5 mM $MgCl_2$, 10% glycerol, 5 mM β-mercaptoethanol (BME), 200 μM ATP) supplemented with 20 mM imidazole and 1 mM phenylmethylsulfonyl fluoride (PMSF) and lysed. The cleared lysate was incubated with $Ni^{2+}$-NTA Agarose beads (Qiagen). Beads were washed with the TRIP13 Purification Buffer supplemented with 50 mM imidazole, and eluted with the TRIP13 Purification Buffer supplemented with 250 mM imidazole. The proteins were incubated with TEV for overnight to remove the His$_6$ tag, and fractionated on a Resource Q column (GE) with the TRIP13 Purification Buffer containing 50 mM NaCl as the starting buffer, and the TRIP13 Purification Buffer containing 1 M NaCl as the elution buffer. TRIP13 proteins were further purified on the HiLoad 16/600 Superdex 200 gel filtration column (GE) in the buffer containing 20 mM HEPES, pH 7.7, 100 mM KCl, 5 mM $MgCl_2$, 5% glycerol, 1 mM DTT. The proteins were concentrated with Centriprep Concentrators (Millipore) and stored at −80 °C.

His$_6$-ΔN35-p31[comet]–Mad2[L13A] complex and His$_6$-Mad2 (containing the C79S and C106S mutations to avoid disulfide formation in vitro) were expressed in BL21 (DE3) cells. Collected cells were lysed in the Mad2 Purification Buffer (50 mM Tris, pH 8.0, 300 mM NaCl, 5 mM BME) supplemented with 1 mM PMSF. The cleared lysates were incubated with $Ni^{2+}$-NTA Agarose beads (Qiagen). Beads were washed

with the Mad2 Purification Buffer supplemented with 10 mM imidazole, and with the MOPS Buffer (50 mM MOPS pH 7.2, 300 mM NaCl, 5 mM BME) supplemented with 25 mM imidazole. Proteins were eluted with the MOPS Buffer supplemented with 200 mM imidazole, and exchanged into the MOPS Buffer with no imidazole with a PD-10 desalting columns (GE). The His$_6$-tag of the $\Delta$N35-p31$^{comet}$–Mad2$^{L13A}$ complex was cleaved by TEV, whereas His$_6$-Mad2 was not cleaved. The proteins were purified with the Resource Q column with the MOPS Buffer containing 75 mM NaCl as the binding buffer, and the MOPS Buffer containing 1 M NaCl as the elution buffer. The proteins were concentrated and further purified with the Superdex 200 Increase 10/300 GL column (GE) in a buffer containing 25 mM Tris, pH 7.5, 200 mM NaCl, 10% glycerol, and 1 mM Tris(2-carboxyethyl)phosphine (TCEP).

His$_6$-$\Delta$N35-p31$^{comet}$ was expressed in M15[pRep4] E. coli cells. Cells were lysed in the buffer containing 50 mM MOPS, pH 7.2, 150 mM NaCl, 5 mM BME, 10 mM imidazole. After purification with Ni$^{2+}$-NTA Agarose beads and TEV cleavage, the protein was further purified with the Resource Q column with the binding buffer of 20 mM HEPES, pH 7.5, 50 mM NaCl, 10% glycerol, and 1 mM DTT and the elution buffer of 20 mM HEPES, pH 7.5, 1 M NaCl, 10% glycerol, and 1 mM DTT. The protein was further purified by gel filtration in the buffer containing 20 mM HEPES, pH 7.5, 200 mM NaCl, 10% glycerol, and 1 mM DTT.

His$_6$-Mad2$^{R133A}$ was expressed in M15[pRep4] cells in M9 minimal media containing $^{15}$N-NH$_4$Cl as the sole nitrogen source. The cells were collected in the R133A Buffer (50 mM sodium phosphate, pH 8.0, 300 mM NaCl, 5 mM BME) supplemented with 10 mM imidazole, lysed by sonication, and cleared by centrifugation. The lysate was incubated with Ni$^{2+}$-NTA Agarose beads (Qiagen). Beads were washed with the R133A Buffer supplemented with 10 mM imidazole, followed by the R133A Buffer supplemented with 20 mM imidazole. The protein was eluted with the R133A Buffer supplemented with 250 mM imidazole. The protein was dialyzed into the buffer containing 50 mM Tris, pH 8.0, 100 mM NaCl, 1 mM DTT, and cleaved overnight with TEV to remove the His$_6$-tag. The cleaved protein was purified with the Resource Q column (GE) with the binding buffer of 20 mM Tris, pH 8.0, 50 mM NaCl, and 1 mM DTT, and the elution buffer of 20 mM Tris, pH 8.0, 1 M NaCl, and 1 mM DTT. The protein was further purified with the Superdex 75 10/300 GL column (GE) in a buffer containing 50 mM sodium phosphate, pH 6.8, 300 mM KCl, and 1 mM DTT.

Expression of Cdc20N-His$_6$ (residues 1–170 of human Cdc20) in BL21 (DE3) cells were induced with 1 mM IPTG for 6 h at 37 °C. The collected cells were lysed in 8 M urea, 100 mM NaH$_2$PO$_4$, and 10 mM Tris at pH 8.0 for overnight at room temperature. The lysate was sonicated, cleared by centrifugation, and incubated with Ni$^{2+}$-NTA Agarose beads (Qiagen). The beads were washed with 8 M urea, 100 mM NaH$_2$PO$_4$, and 10 mM Tris at pH 6.3, and eluted with 8 M urea, 100 mM NaH$_2$PO$_4$, and 10 mM Tris at pH 4.5. The protein was dialyzed into the buffer containing 25 mM Tris, pH 8.0, 300 mM NaCl, 5% glycerol, and 1 mM BME.

**NMR spectroscopy.** All NMR experiments were recorded on Agilent DD2 600 MHz spectrometer at 30 °C. To obtain uniformly $^{15}$N-labeled Mad2$^{R133A}$, we used M9 minimal media with $^{15}$NH$_4$Cl as the sole nitrogen source (1 g l$^{-1}$). Purified $^{15}$N-Mad2 R133A was incubated overnight at room temperature before the NMR experiments to ensure that the two forms of Mad2 have reached equilibrium[13]. The final NMR samples contained 140 µM $^{15}$N-Mad2$^{R133A}$ (after overnight pre-incubation) with or without 4.3 µM $\Delta$N35-p31$^{comet}$ and 0.43 µM TRIP13 dissolved in a buffer containing 25 mM Tris, pH 7.5, 300 mM KCl, 1 mM MgCl$_2$, 1 mM DTT and 5% D$_2$O. ATP was added to the $^{15}$N-Mad2$^{R133A}$–$\Delta$N35-p31$^{comet}$–TRIP13 sample at a final concentration of 1 mM. For monitoring the conformational change of Mad2, a series of 2D $^1$H–$^{15}$N HSQC spectra each lasting 2 h 34 min were acquired for a total acquisition time of 40 h.

For $^1$H/$^2$H (H/D) exchange experiments, the final NMR samples were lyophilized and dissolved in D$_2$O, and contained 150 µM $^{15}$N-Mad2$^{R133A}$ in a buffer containing 25 mM Tris, pH 7.5, 200 mM NaCl, 5 mM MgCl$_2$, and 1 mM DTT. After the acquisition of a 2D $^1$H–$^{15}$N HSQC spectrum lasting 2 h 34 min, ATP (0.1 M in D$_2$O) and the TRIP13–$\Delta$N35-p31$^{comet}$ complex were added into the Mad2$^{R133A}$ D$_2$O sample to final concentrations of 5 mM and 1.5 µM, respectively. A series of 2D $^1$H–$^{15}$N HSQC spectra each lasting 2 h 34 min were acquired for a total acquisition time of 24 h. All data were processed with NMRpipe[68] and analyzed with NMRView[69].

**Crystallization, data collection, and structure determination.** His$_6$-TRIP13$^{E253A}$ $\Delta$C2 (with the C-terminal two residues deleted) was expressed in BL21 (DE3) cells, and purified using Ni$^{2+}$-NTA Agarose resin pre-equilibrated with the lysis buffer (50 mM phosphate, pH 7.5, 300 mM NaCl, 40 mM imidazole, and 5 mM BME). After cleavage by TEV to remove the His$_6$-tag, the TRIP13$^{E253A}$ protein was further purified with a resource Q column followed by a Superdex 200 gel filtration column equilibrated with a buffer containing 20 mM phosphate, pH 8.0, 300 mM NaCl, and 0.5 mM TCEP. The purified protein was concentrated to 2 mg ml$^{-1}$. The $\Delta$N35-p31$^{comet}$–Mad2$^{L13A}$ complex was expressed and purified as previously described[45]. TRIP13$^{E253A}$ and $\Delta$N35-p31$^{comet}$–Mad2$^{L13A}$ were mixed at 1:2 molar ratio in the presence of 200 µM ATP. The protein mixture was fractionated on a Superdex 200 column equilibrated with a buffer containing 20 mM Tris, pH 8.0, 100 mM NaCl, and 0.5 mM TCEP. The TRIP13$^{E253A}$–$\Delta$N35-p31$^{comet}$–Mad2$^{L13A}$ complex was then concentrated to 5.5 mg ml$^{-1}$, and subjected

to crystallization by the sitting-drop vapor diffusion method at 20 °C. The best crystals were obtained with a reservoir solution containing 0.1 M HEPES, pH 7.5, 0.2 M NaCl, and 4% (v/v) isopropanol. Crystals were cryo-protected by increasing the final concentration of glycerol to 30% (v/v) in a step-wise fashion and flash-cooled in liquid nitrogen.

The X-ray diffraction data were collected at beamline 19-ID (SBC-CAT) at the Advanced Photon Source (Argonne National Laboratory, Argonne, Illinois, USA). The data were processed in the program HKL3000[70]. Initial phases were obtained by molecular replacement in the program Phaser[71], using the structure of C. elegans PCH-2 as a search model (PDB ID: 4XGU). Iterative model building and refinement were performed in the programs Coot[72] and PHENIX[73], respectively. Although the TRIP13$^{E253A}$–$\Delta$N35-p31$^{comet}$–Mad2$^{L13A}$ complex was used for crystallization in the presence of ATP, the final crystals only contained TRIP13$^{E253A}$ bound to ADP. The data collection and refinement statistics are summarized in Supplementary Table 1.

**ATPase assays.** All reactions were performed in white OptiPlate-384 plates (Perkin Elmer) at room temperature. TRIP13 and $\Delta$N35-p31$^{comet}$–Mad2$^{L13A}$ were diluted in the Reaction Buffer (25 mM Tris, pH 7.5, 200 mM NaCl, 10 mM MgCl$_2$, 1 mM DTT, 5% glycerol, and 0.05% Tween) to a final volume of 5 µl, and incubated for 15 min. ATP was added to the reaction to a final volume of 10 µl and a final concentration of 0.1 mM, and incubated for 1 h. Volume of 10 µl Glo Reagent from the ADP Glo Kinase Assay kit (Promega) was added to the reaction, and incubated for 40 min. Volume of 20 µl Detect reagent from the kit was added to the reaction, and incubated for 30 min. Luminescence was detected with the VICTOR 3 V Multilabel Plate Reader (Perkin Elmer). A standard curve was prepared according to the ADP Glo kit instructions to calculate the concentration of ADP based on luminescence. All data were normalized to the reading of the sample containing the Reaction Buffer alone with no proteins. The best results were obtained when the ADP Glo kit reagents were diluted twofold with the Reaction Buffer. Different standard curves were used for undiluted vs. diluted reagents.

**Mad2 dissociation and binding assays.** The MBP1 peptide (with the sequence of SWYSYPPPQRAVC) was synthesized at the Protein Chemistry Technology Core at University of Texas Southwestern Medical Center (UTSW). MBP1 or Cdc20N was coupled to SulfoLink Coupling Resin (Thermo Fisher) according to the manufacturer's instructions. Beads were washed in the ATPase Reaction Buffer. His$_6$-Mad2 (0.5 mg ml$^{-1}$) was incubated with the beads for 30–60 min. After washes with the ATPase Reaction Buffer, TRIP13 and $\Delta$N35-p31$^{comet}$ at indicated concentrations and 1 mM ATP were added to the beads and incubated for another 30–60 min. Proteins bound to beads were dissolved in SDS sample buffer, analyzed by SDS-PAGE, and stained with Coomassie blue. The Mad2-binding assay was performed in a similar manner except that ATPγS was used in place of ATP. All reactions were performed in the Reaction Buffer at room temperature.

**Analytical ultracentrifugation.** All analytical ultracentrifugation (AUC) experiments were performed in the sedimentation velocity (SV) mode at 4 °C in an Optima XL-I ultracentrifuge (Beckman-Coulter, Indianapolis, IN). Samples (400 µl) were introduced into the sample sectors of chilled, charcoal-filled dual-sector Epon centerpieces that had been sandwiched between sapphire windows in standard AUC cell housings. An equal volume of AUC buffer (10 mM HEPES, pH 7.4, 300 mM NaCl, 5 mM MgCl$_2$, 1.357 M glycerol, and 1 mM TCEP) was placed in the reference sectors. One sample of monomeric bovine serum albumen (BSA, Sigma-Aldrich, St. Louis, MO) was also centrifuged to serve as a molar-mass standard, and to determine whether the calculated (see below) solution parameters were realistic. BSA had been purified over a Superdex 200 16/60 size-exclusion. When present, ATP or ATPγS was included at a concentration of 100 µM in both the sample and reference buffers. The assembled cells were positioned in a chilled An50-Ti rotor, which was then put into the centrifuge and incubated at the experimental temperature under vacuum overnight. Next, the rotor was accelerated to 201,600 g and concentration profiles were collected using the onboard absorbance optics tuned to 280 nm. The data were collected until there was no evidence of sedimentation. Solution parameters (density, viscosity) were calculated in SEDNTERP[74]. Using SEDFIT[75] and the c(s) methodology, we calculated the molar mass of BSA, which was within 10% of the known value (66,430 Da). Thus, the calculated solution parameters were deemed acceptable, and SEDFIT was used to analyze the TRIP13 samples. Maximum-entropy regularization with a confidence level of 0.683 was used, and time-invariant noise elements in the data were subtracted[76]. An s-resolution of 100 was used. All sedimentation coefficients were converted to standard conditions (water, 20 °C) using GUSSI[77].

**APC/C ubiquitination assay.** BubR1N (residues 1–370), His$_6$-Cdc20 full-length, and dimeric Mad2 used in the APC/C assays were purified as described[29]. Uba1, UbcH10, and securin-Myc were purified as described[78]. Ube2S was a gift from Dr. Michael Rape (University of California at Berkeley). APC/C was isolated from Xenopus egg extracts with anti-APC3 antibody-coupled Protein A beads (Bio-Rad). Each reaction contained 5 µl of beads that were incubated with 50 µl of extract. The beads were supplemented with 3.8 µM His$_6$-Cdc20 for 1 h and washed. The minimal functional MCC was formed by an incubation of BubR1N, Cdc20, and

Mad2 for 20 min. APC/C$^{Cdc20}$ beads were incubated with the pre-formed MCC and washed. The ubiquitination reaction was performed in the presence of securin-Myc for 30 min, and quenched with the SDS loading buffer. The reaction mixtures were analyzed by SDS-PAGE followed by immunoblotting with the anti-Myc antibody (monoclonal; clone 9E10).

For TRIP13 inactivation of MCC, TRIP13 was diluted and mixed with p31$^{comet}$ and added to the MCC mixture in the presence of 5 mM MgCl$_2$ and 1 mM ATP. The final reaction volume was 8 µl. This reaction was incubated for 1 h, and diluted to a final volume of 25 µl by the XB buffer (10 mM HEPES, pH 7.7, 100 mM KCl, 0.1 mM CaCl$_2$, 1 mM MgCl$_2$, and 50 mM sucrose) supplemented with 2 mg ml$^{-1}$ BSA. The APC/C$^{Cdc20}$ beads (5 µl) were then added, and the APC/C ubiquitination assay was performed. In the case of MCC bound to APC/C, the MCC was first incubated with the APC/C$^{Cdc20}$ beads, and then the TRIP13 reaction was performed.

**Data availability**. The coordinates of the human TRIP13–ADP structure have been deposited to the Protein Data Bank with the PDB ID: 5WC2. All other relevant data are available upon request from the authors.

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

## Acknowledgements

We thank Haydn Ball at the UTSW Protein Chemistry Technology Core for peptide synthesis, Shih-Chia Tso in the UTSW Macromolecular Biophysics Resource for assistance with analytical ultracentrifugation, and Michael Rape at the University of California at Berkeley for providing recombinant Ube2S. Use of Argonne National Laboratory Structural Biology Center beamlines at the Advanced Photon Source was supported by the US Department of Energy under contract DE-AC02-06CH11357. This study is supported by grants from the National Institutes of Health (GM107415 to X.L.), Cancer Prevention and Research Institute of Texas (RP160255 to X.L.) and the Welch Foundation (I-1932 to X.L. and I-1441 to H.Y.). H.Y. is an Investigator with the Howard Hughes Medical Institute.

## Author contributions

M.L.B. conceived the project, expressed and purified TRIP13 proteins, performed all ATPase, Mad2 dissociation, and protein-binding assays, and wrote the initial draft of the paper. B.-C.J. performed crystallization and structure determination of human TRIP13. F.L. performed the NMR and H/D exchange experiments. B.L. performed the APC/C ubiquitination assays. E.B.Y. purified [15]N-labeled Mad2 and performed the initial NMR experiment. Q.W. assisted in the acquisition and analysis of NMR data. C.A.B. performed the analytical ultracentrifugation experiments. H.Y. and X.L. supervised the project, analyzed the results, and wrote the paper with input from all authors.

## Additional information

**Competing interests:** The authors declare no competing financial interests.

