## [Peer Review File · Nature Communications]

Reviewers' comments:

Reviewer #1 (Remarks to the Author):

In the manuscript “Mechanistic Insight into TRIP13-catalyzed Mad2 Structural Transition and Spindle Checkpoint Silencing, Brulotte et al show a nice structural and biochemical dissection of the role of TRIP13 and p31comet in the spindle assembly checkpoint. Briefly, they determine the crystal structure of TRIP13, identify mutants that affect ATP hydrolysis, substrate binding, or both, and the very nicely demonstrate that TRIP13 and p31comet, together, can disassemble functional mitotic checkpoint complexes and enable APC/C-Cdc20 activation.

This work is important, well-executed, and explained well. I support eventual publication of this manuscript in Nature Communications, pending revision to address the concerns below.

My principal concern with this manuscript is that it does not cite or discuss a very recent paper published in EMBO Journal by the Corbett laboratory (<https://www.ncbi.nlm.nih.gov/pubmed/28659378>), which demonstrated many of these findings already. In particular, they solved the crystal structure of TRIP13, and also demonstrated that TRIP13 functions by engaging the N-terminus of MAD2 and partially unfolding this segment to mediate C-MAD2 to O-MAD2 conformational conversion. The current manuscript provides valuable new insights, so its overlap with the Corbett paper does not significantly lessen its potential impact; however, the authors should address this work and how it fits with their new findings throughout the manuscript.

Particular points:

Introduction, page 4: The term “Intermediate” or “I-MAD2” has been used by the authors and other groups previously to mean either the transient partially-unfolded transition state adopted during O-MAD2 to C-MAD2 conversion, or more recently in Hara et al PNAS 2015, as a fully-folded state that has O-MAD2 topology but a core-fold conformation closer to C-MAD2. The authors should consider more clearly explaining which of these they mean by “I-MAD2” in the introduction.

Introduction, page 4: The idea that p31comet binds MAD1:MAD2 at kinetochores and inactivates its ability to convert more O-MAD2 to C-MAD2 is a long-standing one, but it’s problematic. Why would it only do this upon SAC activation, as the authors strongly imply? I

believe p31comet (and indeed TRIP13 as well) is localized at unattached kinetochores even when the SAC is active. The authors may want to more clearly address this point.

Introduction, page 4-5: The recent Corbett group paper should be cited and discussed here, as it provides significant new insight into the physical mechanisms of p31comet and TRIP13.

Results, page 6: The NMR experiments showing MAD2 conversion are very nice. Is it possible to do a time-course analysis, of the type the authors have previously done to examine spontaneous O-MAD2 to C-MAD2 conversion, but this time to measure the rate of TRIP13-mediated conversion?

Results, page 7: Question: If TRIP13 were to mediate a fast conversion between C-MAD2 and O-MAD2 that DID involve transient complete unfolding of MAD2, would the amides in question be exchanged in measurable quantities? That is, do these experiments have the ability to detect a single transient unfolding event of the type the authors are trying to disprove with this experiment?

Also, for the second set of HDX experiments, I would also like to see the results mapped onto the structure of O-MAD2.

Also, the Corbett paper suggests that TRIP13 unfolds the N-terminus and alpha-A helix of MAD2; can the authors meaningfully comment on this model, given their HD exchange data?

Figure 2 - panels B and C are inconsistently labeled in the figure and the legend.

Results, page 8-9: The authors refer to their crystal structure of TRIP13 as “monomeric”. The crystal form appears to be the same as that reported by the Corbett group (PDB codes 5VQ9 and 5VQA). They describe a crystal in which the crystallographic asymmetric unit contains a dimer, but that symmetry operators recapitulate a biologically-relevant subunit-subunit interface. The authors should refer to this previous work, and amend their description of “monomeric” TRIP13 to better reflect the crystal packing (if indeed theirs is the same as the Corbett group’s). While the Corbett group’s structures are of Apo and ATP-bound TRIP13, the current structure is ADP-bound. As these other structures are available, the authors should compare the states and identify any potential meaningful differences.

Results, page 12 and figure 6: The authors discuss G320 in the manuscript extensively, but do not show its exact location in Figure 6C (III). They should show it.

Results, page 14-15: In their supplemental data, the Corbett group showed gel-filtration profiles of TRIP13 and several mutants in the presence and absence of nucleotides. The authors should note this work and compare/contrast their findings with regard to TRIP13 oligomerization.

Results, page 16, paragraph 1: This is a rather confusing discussion of N278's position in different protomers of the TRIP13 hexamer. Are the authors referring the PCH-2 hexamer structure here, or a TRIP13 hexamer model based on that structure? As the authors note, they don't have the TRIP13 hexamer structure.

Results, page 16, paragraph 2: "dissembles" should be "disassembles"

Results, page 16-17: the MCC disassembly assay and the failure of p31comet and TRIP13 to re-activate APC/C that has already been inhibited is really interesting and nicely done. I would like to see whether truncating the N-terminus of MAD2 (as done in the Corbett group's EMBO Journal paper) would disrupt this activity.

Discussion: The authors' model of MAD2 C-terminus unfolding by TRIP13 makes perfect sense in light of their own data and the known structures of C-MAD2 and O-MAD2, but runs counter to the Corbett group's model (their model of N-terminus unfolding is counter-intuitive when considering MAD2 structure, but is consistent with their data). The authors should at the very least address this discrepancy between the two models when discussing TRIP13's likely mechanism.

Reviewer #2 (Remarks to the Author):

The paper describes a collection of results from bio-physical and –chemical experiments that elucidate the mechanism of closed-to-open state transition of Mad2 catalyzed by TRIP13. This is then related to the role of TRIP13 in spindle checkpoint silencing. Given the high significance of correct spindle checkpoint silencing in chromosome segregation and therefore also in many diseases, detailed understanding of the mechanism is important.

In detail, the findings are:

- 1) TRIP13 + p31comet catalyze Mad2 opening without disrupting its core fold, driven by ATP hydrolysis.
- 2) Determination of the X-ray structure bound to ADP (rather than ATP).
- 3) TRIP13 + p31comet reverse inhibition of APC/C via Mad2.
- 4) Assays identify the role of specific TRIP13 residues in ATP hydrolysis or binding to p31comet.

Taken together, the paper provides a deepened and interesting insight into the investigated mechanisms. In particular, findings 3 and 4 are valuable.

On the other hand, the NMR investigations are somewhat underwhelming. NMR spectroscopy demonstrates that TRIP13 indeed catalyzes the Mad2 rearrangement, which has already been

shown previously, see ref. 69. The only new finding is that the unfolding takes place without disruption of the core of Mad2. This finding is convincing thanks to tracing H/D exchange. It appears that the authors did not expect this. Why so? I would have been surprised if complete unfolding had been involved. Is it because ClpX involves complete unfolding? What molecule does ClpX unfold? Is it comparable to Mad2? This should be discussed.

There is not really a structural picture of the mechanism emerging (the authors promise “key insight into the mechanism of TRIP13-catalyzed conformational change of Mad2”). It seems relatively easy to collect much more information from NMR. For example, the X-ray attempt to solve the structure of the complex failed. Why not checking the involved residues by NMR such that a crude complex model could be provided nevertheless? Simple chemical shift mapping of the Mad2 peaks upon TRIP13+p31 comet titration would show where the binding hot spots are. As the reaction takes hours it would also be interesting to observe the changes of the peaks over time in HSQC spectra.

Therefore, the new insights from a structural point of view are somewhat incremental. The NMR investigation stopped where it would turn interesting.

I also wonder if a NMR analysis of the TRIP13 is possible. Could TRIP13 be studied in its monomeric form (X-ray suggests so)? If so, chemical shift analysis could be repeated on TRIP13 and extend the insight considerably. It may even be possible to conduct CPMG experiments to substantiate the repeatedly mentioned conformational exchange dynamics. (However, this would be clearly beyond the scope of this paper.)

The following points should also be addressed:

-The NMR proof of the Mad2 conversion upon addition of TRIP13 is based on a HSQC spectrum before and after addition of TRIP13, which exhibits the peaks expected for the C- or O-conformation. What samples were used for the reference spectra (magenta and blue in figure 1b)? Apparently, there is another way to induce the conformational switch.

-What is rational behind the choice of the mutants? Since the complex structure is not known, the optimal choice is not that obvious.

-The pore loop should be indicated in figure 3, which would help to follow the extensive discussion on page 9.

-Why is the activity in figures S2 c) and d) not the same for the wild type?

-An interesting finding is that the G320A mutation has a major impact. While it is not unreasonable to assume that the mutation affects dynamics, I would tone down that claim in the title on page 13. It would also be interesting to see a close-up of the hinge loop including side

chains. It is not clear if residue 320 is located at the hinge or in the middle of the loop.

-It is not clear to me how the ATPase activity is coupled to the transition. Could the authors add one or two phrases explaining this?

Based on the overall significance and novelty of the gained insights, and the rigorousness of the conclusions, I recommend the paper be published in Nature Communications if my points can be addressed to my satisfaction.

Minor:

p. 5: "...local unfolding of minimally the C-terminal region" should be corrected.

Figure caption 1: C-Mad2, ¹⁵N-Mad2: both are ¹⁵N labeled, as far as I understand, I don't understand the nomenclature

p. 7: "...slowing-exchanging..."

Figure caption 2: b) and c) should be switched

Reviewer #1 (Remarks to the Author):

My principal concern with this manuscript is that it does not cite or discuss a very recent paper published in EMBO Journal by the Corbett laboratory (<https://www.ncbi.nlm.nih.gov/pubmed/28659378>), which demonstrated many of these findings already. In particular, they solved the crystal structure of TRIP13, and also demonstrated that TRIP13 functions by engaging the N-terminus of MAD2 and partially unfolding this segment to mediate C-MAD2 to O-MAD2 conformational conversion. The current manuscript provides valuable new insights, so its overlap with the Corbett paper does not significantly lessen its potential impact; however, the authors should address this work and how it fits with their new findings throughout the manuscript.

Response: We appreciate the reviewer's positive assessment of our work. The Corbett study was published online near the time when we submitted our manuscript. We did not have a chance to cite and discuss that study at the time. We have cited the Corbett study and compared and contrasted our results with theirs in the revised manuscript (see also responses to specific points below).

Particular points:

Introduction, page 4: The term "Intermediate" or "I-MAD2" has been used by the authors and other groups previously to mean either the transient partially-unfolded transition state adopted during O-MAD2 to C-MAD2 conversion, or more recently in Hara et al PNAS 2015, as a fully-folded state that has O-MAD2 topology but a core-fold conformation closer to C-MAD2. The authors should consider more clearly explaining which of these they mean by "I-MAD2" in the introduction.

Response: We and the Musacchio group have agreed to term the most populated state of the Mad2 copy bound to C-Mad2 as I-Mad2. Because the NMR spectra of I-Mad2 were very different from those of O-Mad2, we initially thought that I-Mad2 might be locally unfolded in its C-terminal region. Structure determination of I-Mad2 revealed that it is still a fully folded state with O-Mad2 topology. When we describe I-Mad2, we are referring to the most populated conformation of Mad2 bound to C-Mad2, first determined by the Musacchio group and further clarified in the Hara *et al.* PNAS paper. To avoid confusion, we have modified the text as follows: "The Mad1-C-Mad2 complex at kinetochores recruits additional O-Mad2 from the cytosol and converts it to an intermediate form (I-Mad2)^{21,52,53}. Through a high-energy transition state in which its C-terminal region is locally unfolded, I-Mad2 binds to Cdc20 to form the Cdc20-C-Mad2 complex, which then associates with BubR1-Bub3 to produce MCC⁵³⁻⁵⁵".

Introduction, page 4: The idea that p31^{comet} binds MAD1:MAD2 at kinetochores and inactivates its ability to convert more O-MAD2 to C-MAD2 is a long-standing one, but it's problematic. Why would it only do this upon SAC (in)activation, as the authors strongly imply? I believe p31^{comet} (and indeed TRIP13 as well) is localized at unattached kinetochores even when the SAC is active. The authors may want to more clearly address this point.

Response: We fully agree with the reviewer. We also observe p31^{comet} at kinetochores when the checkpoint is active. There must be Mad1-C-Mad2 core complexes not capped by p31^{comet} at unattached kinetochores that catalyze Mad2 conformational activation, however. There is

currently no evidence that p31^{comet} is regulated. We have modified our statement as follows: “The C-Mad2-binding protein p31^{comet} binds to the Mad1–C-Mad2 core and limits Mad2 conformational activation, at kinetochores when the checkpoint is on and in the cytosol after Mad1–C-Mad2 is removed from kinetochores during checkpoint inactivation⁵⁶⁻⁵⁹”.

Introduction, page 4-5: The recent Corbett group paper should be cited and discussed here, as it provides significant new insight into the physical mechanisms of p31comet and TRIP13.

Response: We have cited and discussed the Corbett study in the Introduction of the revised manuscript.

Results, page 6: The NMR experiments showing MAD2 conversion are very nice. Is it possible to do a time-course analysis, of the type the authors have previously done to examine spontaneous O-MAD2 to C-MAD2 conversion, but this time to measure the rate of TRIP13-mediated conversion?

Response: This is a great suggestion. We had indeed performed a time course experiment. The acquisition of each HSQC spectrum took 2.5 hrs. The third HSQC spectrum in this time series was shown in Fig. 1c. We could only detect about 5-10% C-Mad2 in the first HSQC spectrum after TRIP13 and ATP addition. Thus, we could not obtain enough timepoints to calculate the rate. In addition, the determination of apparent k_{cat} and K_m would require varying the concentrations of TRIP13. Thus, analysis of kinetics of TRIP13-catalyzed Mad2 conformational change requires the development of faster biophysical assays (e.g. FRET) and preferably at the single-molecule level.

Results, page 7: Question: If TRIP13 were to mediate a fast conversion between C-MAD2 and O-MAD2 that DID involve transient complete unfolding of MAD2, would the amides in question be exchanged in measurable quantities? That is, do these experiments have the ability to detect a single transient unfolding event of the type the authors are trying to disprove with this experiment?

Response: The H/D exchange of amide protons in dynamic unstructured regions happens on the millisecond-second timescale. The complete unfolding of a stable substrate by the analogous ClpXP system occurs at a rate of 0.9 min^{-1} (Kim et al. *Mol. Cell* 4, 639-648, 2000). Even though we could not measure the rate of Mad2 conformational switch catalyzed by TRIP13, we could observe about 5-10% of C-Mad2 in the first HSQC spectrum that took 2.5 hrs to acquire. This finding suggested that TRIP13 (at 1:325 molar ratio) have not finished converting all C-Mad2 to O-Mad2 in 150 minutes. We estimate that TRIP13-catalyzed Mad2 conformational transition occurs at a timescale of tens of seconds, similar to that of ClpXP-catalyzed substrate unfolding. Thus, the TRIP13-catalyzed Mad2 conformational transition is a slow process, much slower than H/D exchange of amide protons in unstructured regions at neutral pH. If the Mad2 transition catalyzed by TRIP13 were to occur through global unfolding, the amide protons in its stably folded core would have had enough time to exchange with the D₂O solvent and become undetectable. Unfortunately, the paper is already too long (the journal asks that the main text be under 5,000 words). There is no space to add this discussion to the text. We hope that the reviewer would understand.

Also, for the second set of HDX experiments, I would also like to see the results mapped onto the structure of O-MAD2.

Response: This is a great suggestion. We have done so, and the results are included in Fig. 2e.

Also, the Corbett paper suggests that TRIP13 unfolds the N-terminus and alpha-A helix of MAD2; can the authors meaningfully comment on this model, given their HD exchange data?

Response: The most crucial evidence for Mad2 N-terminal unfolding by TRIP13 in the Corbett paper is the crosslinking data performed with the TRIP13 W221C mutant. This crosslinking appeared to show that the N-terminal region of Mad2 directly engages the pore loop of TRIP13, which is further inferred as evidence for the threading of this region of Mad2 into the central pore. On the other hand, we have shown that the W221A mutation abrogates binding of TRIP13 to p31^{comet}-Mad2 in the presence of ATPγS (Fig. 6a and Supplementary Fig. 6b). We wondered whether the W221C mutant used in the Corbett study was similarly defective in binding to p31^{comet}-Mad2. We thus made the TRIP13 W221C mutant (Supplementary Fig. 6c). Expectedly, the W221C mutant behaved very similarly to the W221A mutant. It had higher basal ATPase activity, which was not further stimulated by p31^{comet}-Mad2 (Supplementary Fig. 6d). W221C also could not dissociate Mad2 from its ligand MBP1 (Supplementary Fig. 6e). More importantly, similar to W221A, W221C lost its binding to p31^{comet}-Mad2 (Supplementary Fig. 6f). Therefore, the installation of a crosslinkable residue in the pore loop destroys its function. Results obtained from crosslinking experiments with a non-functional, Mad2-binding-deficient mutant of TRIP13 are unlikely to be valid. Evidence supporting the local unfolding of Mad2 from its N-terminus by TRIP13 is thus lacking. Our results do not definitively show from which terminus Mad2 unfolding initiates. Because the C-terminal region of Mad2 needs to be remodeled, the model involving local unfolding of Mad2 C-terminal region by TRIP13 makes more intuitive sense, as the reviewer pointed out.

Figure 2 - panels B and C are inconsistently labeled in the figure and the legend.

Response: The error has been corrected.

Results, page 8-9: The authors refer to their crystal structure of TRIP13 as “monomeric”. The crystal form appears to be the same as that reported by the Corbett group (PDB codes 5VQ9 and 5VQA). They describe a crystal in which the crystallographic asymmetric unit contains a dimer, but that symmetry operators recapitulate a biologically-relevant subunit-subunit interface. The authors should refer to this previous work, and amend their description of “monomeric” TRIP13 to better reflect the crystal packing (if indeed theirs is the same as the Corbett group’s). While the Corbett group’s structures are of Apo and ATP-bound TRIP13, the current structure is ADP-bound. As these other structures are available, the authors should compare the states and identify any potential meaningful differences.

Response: Again, this is a great suggestion. The space group of our crystals is indeed the same as that of the Corbett study. In both cases, each asymmetric unit contains only one TRIP13 molecule. We believe that TRIP13 is likely monomeric in the crystal, even though it is a

hexamer in solution. Our structure of human TRIP13 bound to ADP is virtually identical to those of human apo-TRIP13 or TRIP13 bound to ATP, determined by the Corbett group (Supplementary Fig. 2a). All three structures are in the “closed” state. Thus, nucleotide binding alone is insufficient to trigger conformational changes in TRIP13 in the monomer state. As noted for apo-TRIP13 and ATP-bound TRIP13, ADP-bound TRIP13 also forms a helical filament in crystal packing (Supplementary Fig. 2b). The interface between TRIP13 monomers in this crystal contact closely resembles that of the interface between two closed protomers in PCH-2 (Supplementary Fig. 2b). The crystal contacts do not recapitulate the interface between closed and open protomers of the PCH-2 hexamer, however. Moreover, the human TRIP13 structure has a disordered pore loop. We thus used the structure of the *C. elegans* PCH-2 hexamer to guide our subsequent mutagenesis and biochemical studies. We have added discussion along this line in the revised manuscript.

Results, page 12 and figure 6: The authors discuss G320 in the manuscript extensively, but do not show its exact location in Figure 6C (III). They should show it.

Response: The position of G320 is now shown in Fig. 5d and 6c.

Results, page 14-15: In their supplemental data, the Corbett group showed gel-filtration profiles of TRIP13 and several mutants in the presence and absence of nucleotides. The authors should note this work and compare/contrast their findings with regard to TRIP13 oligomerization.

Response: The Corbett group showed that TRIP13 WT formed a mixture of hexamers, trimers, dimers, and monomers in the presence of ATP on gel filtration. TRIP13 ATPase-dead mutants form more discrete hexamers. Our results on TRIP13 WT are very similar to theirs. They did not examine the G320A mutant. We have added discussion along this line in the revised manuscript.

Results, page 16, paragraph 1: This is a rather confusing discussion of N278’s position in different protomers of the TRIP13 hexamer. Are the authors referring the PCH-2 hexamer structure here, or a TRIP13 hexamer model based on that structure? As the authors note, they don’t have the TRIP13 hexamer structure.

Response: We have clarified this point. The structure shown is that of the PCH-2 hexamer.

Results, page 16, paragraph 2: “dissembles” should be “disassembles”

Response: The typo has been corrected.

Results, page 16-17: the MCC disassembly assay and the failure of p31^{comet} and TRIP13 to re-activate APC/C that has already been inhibited is really interesting and nicely done. I would like to see whether truncating the N-terminus of MAD2 (as done in the Corbett group’s EMBO Journal paper) would disrupt this activity.

Response: This is a great suggestion. We have performed the experiment with Δ N10-Mad2 (replacing the first 10 residues of Mad2 with 12 residues from an artificial 12-residue tag, as we

have described previously). Mad2 Δ N10 could be incorporated into a functional MCC that inhibited APC/C^{Cdc20} (Fig. 7c). In contrast to MCC containing Mad2 WT, MCC containing Mad2 Δ N10 was not efficiently inactivated by TRIP13, and APC/C activity was not fully restored. Furthermore, Mad2 Δ N10 retained its binding to p31^{comet}, but the Mad2 Δ N10-p31^{comet} complex was deficient in TRIP13 binding (Supplementary Fig. 6g). Therefore, consistent with the results of Corbett and coworkers, the N-terminal region of Mad2 is required for TRIP13 binding and proper MCC disassembly. In other words, the conclusion by Corbett's group that the N-terminus of Mad2 mediates TRIP13 binding is correct, but one cannot know whether this region directly engages the pore loop of TRIP13 and whether Mad2 unfolding initiates from this terminus.

Discussion: The authors' model of MAD2 C-terminus unfolding by TRIP13 makes perfect sense in light of their own data and the known structures of C-MAD2 and O-MAD2, but runs counter to the Corbett group's model (their model of N-terminus unfolding is counter-intuitive when considering MAD2 structure, but is consistent with their data). The authors should at the very least address this discrepancy between the two models when discussing TRIP13's likely mechanism.

Response: The recent Corbett study suggests that the N-terminal region of Mad2 *directly* engages the pore loop of TRIP13 and that Mad2 unfolding by TRIP13 starts at the N-terminus. The key evidence supporting these claims involves crosslinking experiments with the TRIP13 W221C mutant. We have now shown that this pore loop mutant is defective in p31^{comet}-Mad2 binding and in Mad2 unfolding, casting doubt on the validity of the crosslinking data. Therefore, the evidence that supports the *direct* engagement of Mad2 N-terminus with the pore loop of TRIP13 is flawed. On the other hand, consistent with the previously reported cellular data, the Mad2 N-terminus is required for TRIP13 binding (note that this may not be through *direct* engagement with the pore loop) and for TRIP13-catalyzed disassembly of MCC. Although it makes more intuitive sense that TRIP13 initiates Mad2 unfolding locally from the C-terminus, our results do not disprove that TRIP13 unfolds Mad2 from the N-terminus. High-resolution structural analysis of the TRIP13-p31^{comet}-Mad2 complex is needed to definitively settle this issue. We have added this discussion in the revised manuscript.

Reviewer #2 (Remarks to the Author):

The paper describes a collection of results from bio-physical and –chemical experiments that elucidate the mechanism of closed-to-open state transition of Mad2 catalyzed by TRIP13. This is then related to the role of TRIP13 in spindle checkpoint silencing. Given the high significance of correct spindle checkpoint silencing in chromosome segregation and therefore also in many diseases, detailed understanding of the mechanism is important.

In detail, the findings are:

- 1) TRIP13 + p31comet catalyze Mad2 opening without disrupting its core fold, driven by ATP hydrolysis.*
- 2) Determination of the X-ray structure bound to ADP (rather than ATP).*
- 3) TRIP13 + p31comet reverse inhibition of APC/C via Mad2.*
- 4) Assays identify the role of specific TRIP13 residues in ATP hydrolysis or binding to p31comet.*

Taken together, the paper provides a deepened and interesting insight into the investigated mechanisms. In particular, findings 3 and 4 are valuable.

Response: We thank the reviewer for the overall positive comments about our work and for pointing out the importance of our findings.

On the other hand, the NMR investigations are somewhat underwhelming. NMR spectroscopy demonstrates that TRIP13 indeed catalyzes the Mad2 rearrangement, which has already been shown previously, see ref. 69. The only new finding is that the unfolding takes place without disruption of the core of Mad2. This finding is convincing thanks to tracing H/D exchange. It appears that the authors did not expect this. Why so? I would have been surprised if complete unfolding had been involved. Is it because ClpX involves complete unfolding? What molecule does ClpX unfold? Is it comparable to Mad2? This should be discussed.

Response: We agree with the reviewer that our demonstration of TRIP13-catalyzed Mad2 conformational change is not entirely novel, and we did not claim that. The original ref. 69 (now ref. 55) showed that TRIP13 catalyzes Mad2 structural transition by using anion exchange chromatography. We wish to point out that NMR spectroscopy is a more definitive method to demonstrate the conformational change of Mad2. In fact, it was not trivial to find the right conditions to demonstrate this by NMR. In any case, the fact that multiple methods confirm the same finding will be reassuring to the field. With the H/D exchange experiments, we were simply trying to differentiate two possible models of TRIP13-catalyzed Mad2 conformational change: global unfolding versus local unfolding. Before we performed the experiment, we did not favor or disfavor either model. We did not find the local unfolding result particularly unexpected. We have removed the word “Strikingly” from the text to avoid any confusion. The substrates of ClpX are bacterial proteins, which are not structurally related to Mad2. We have added a statement in the Discussion to clarify this.

There is not really a structural picture of the mechanism emerging (the authors promise “key insight into the mechanism of TRIP13-catalyzed conformational change of Mad2”). It seems relatively easy to collect much more information from NMR. For example, the X-ray attempt to solve the structure of the complex failed. Why not checking the involved residues by NMR such that a crude complex model could be provided nevertheless? Simple chemical shift mapping of

the Mad2 peaks upon TRIP13+p31^{comet} titration would show where the binding hot spots are. As the reaction takes hours it would also be interesting to observe the changes of the peaks over time in HSQC spectra. Therefore, the new insights from a structural point of view are somewhat incremental. The NMR investigation stopped where it would turn interesting.

Response: We agree that the crystal structure of the TRIP13-p31^{comet}-Mad2 complex would have been more informative. On the other hand, we feel that our biophysical and biochemical experiments, coupled with extensive mutagenesis, have provided key insight into the mechanism of TRIP13. Note that we have refrained from claiming structural insight. The NMR experiments suggested by the reviewer are, unfortunately, not technically feasible at present. To map chemical shift perturbations of Mad2, we would need to add p31^{comet} and TRIP13 at stoichiometric amounts (as opposed to the 1:33 or 1:325 molar ratios used in the catalysis experiment). We have attempted to acquire the HSQC spectrum (the TROSY version) of the Mad2-p31^{comet} complex, with only Mad2 labeled with ¹⁵N. As shown in **Figure-to-reviewer**, virtually all Mad2 peaks disappeared when it is bound to p31^{comet}. Therefore, the relatively large size of the Mad2-p31^{comet} complex (50 kD) and quite possibly exchange broadening prohibit us from acquiring usable NMR spectra on that complex. The size of the Mad2-p31^{comet}-TRIP13 complex is about 300 kD. NMR studies of such a large complex would be even more challenging. In addition, we cannot concentrate the Mad2-p31^{comet}-TRIP13 complex to concentrations higher than 10 μ M. Therefore, it is impractical to study the complex by NMR spectroscopy. We are, however, attempting to determine the structure of the complex with cryo-electron microscopy. This effort is beyond the scope of the current paper. Unfortunately, the paper is already too long (the journal asks that the main text be under 5,000 words). There is no space to add this discussion to the text. We hope that the reviewer would understand.

Figure-to-reviewer. (a) Coomassie-stained SDS-PAGE gel of the purified recombinant Mad2-p31^{comet} complex that contains ¹⁵N-labeled C-Mad2^{R133A} and unlabeled Δ N35-p31^{comet}. (b) Overlay of the ¹H-¹⁵N TROSY-HSQC spectra of C-Mad2^{R133A} alone (in red) and C-Mad2^{R133A}- Δ N35-p31^{comet} complex (in blue). The sample concentration is 140 μ M of proteins. Each spectrum lasting 24 hours was acquired on an 800 MHz NMR spectrometer.

I also wonder if a NMR analysis of the TRIP13 is possible. Could TRIP13 be studied in its monomeric form (X-ray suggests so)? If so, chemical shift analysis could be repeated on TRIP13

and extend the insight considerably. It may even be possible to conduct CPMG experiments to substantiate the repeatedly mentioned conformational exchange dynamics. (However, this would be clearly beyond the scope of this paper.)

Response: Even though the structure of TRIP13 E253A in the crystal is monomeric, the same protein is hexameric in solution. We currently do not understand why this is. In any case, we do not have a TRIP13 mutant that is monomeric in solution for us to attempt NMR analysis. Moreover, lessons learned from a monomeric mutant of TRIP13 may not be relevant to the functional state of the protein, which is hexameric. If technically feasible, CPMG experiments on the TRIP13 monomer and hexamer might be informative, but this would require considerable effort, including the development of novel methyl labeling schemes and ways to improve the solubility of the protein. As stated by the reviewer, this is clearly beyond the scope of this paper.

The following points should also be addressed:

-The NMR proof of the Mad2 conversion upon addition of TRIP13 is based on a HSQC spectrum before and after addition of TRIP13, which exhibits the peaks expected for the C- or O-conformation. What samples were used for the reference spectra (magenta and blue in figure 1b)? Apparently, there is another way to induce the conformational switch.

Response: As we have shown previously (ref. 13), Mad2 expressed in bacteria has two conformers, which can be separated by anion exchange chromatography: C-Mad2 elutes at higher salt as compared to O-Mad2. This allows us to obtain both forms of Mad2 and acquire reference spectra. C-Mad2 is more thermodynamically stable than O-Mad2. At 30°C, O-Mad2 spontaneously converts to C-Mad2, with a half-life of 9 hours. At low temperature (4°C), this conversion is extremely slow. In the absence of p31^{comet}, TRIP13, and ATP, C-Mad2 does not convert back to O-Mad2.

-What is rational behind the choice of the mutants? Since the complex structure is not known, the optimal choice is not that obvious.

Response: We targeted all TRIP13 residues conserved from yeast to man that are surface exposed or partially exposed in the hexameric *C. elegans* PCH-2 structure. Our rationale is that only surface exposed residues have the chance to form direct contact with p31^{comet}-Mad2, and functionally important residues are often conserved. Not all mutants are presented in the study. We have now clarified this point in the revised manuscript.

-The pore loop should be indicated in figure 3, which would help to follow the extensive discussion on page 9.

Response: This is a great suggestion. We have now labeled the pore loop of PCH-2 in Fig. 3c. The pore loop is missing in the structure of human TRIP13.

-Why is the activity in figures S2 c) and d) not the same for the wild type?

Response: The basal level ATPase activity of TRIP13 is low, and is variable among different TRIP13 preparations. Supplementary Fig. 2c was performed with an earlier preparation of TRIP13, which was less active.

-An interesting finding is that the G320A mutation has a major impact. While it is not unreasonable to assume that the mutation affects dynamics, I would tone down that claim in the title on page 13. It would also be interesting to see a close-up of the hinge loop including side chains. It is not clear if residue 320 is located at the hinge or in the middle of the loop.

Response: We agree with the reviewer. We do not have direct evidence that the hinge actually impacts TRIP13 conformational dynamics. We have changed the title of this section to “**The hinge impacts TRIP13 activity and oligomerization**”, which better reflects our results. We have now shown G320 in sticks in Fig. 5d and 6c. Because G320 has no side chain, we have shown the main chain. G320 is located in the middle of this loop. The loop is very short, and we refer to the entire loop as the hinge.

-It is not clear to me how the ATPase activity is coupled to the transition. Could the authors add one or two phrases explaining this?

Response: We do not know exactly how the ATPase activity is coupled to the structural transition of Mad2. As outlined in Fig. 8a, we propose that the C-terminus of Mad2 engages with the pore loop of TRIP13. Movement of the pore loop driven by ATP hydrolysis tugs on this region of Mad2. The C-terminal region of Mad2 may be threaded into the central pore of TRIP13. Alternatively, the translocation of the Mad2 C-terminus driven by the pore loop may occur exclusively at the top face of TRIP13. In either case, the ATP-driven pore loop movement causes the local unfolding of the Mad2 C-terminal region and its conformational transition to O-Mad2. We have made this point more clearly in the revised manuscript.

Reviewers' comments:

Reviewer #1 (Remarks to the Author):

The revised version of “Mechanistic Insight into TRIP13-catalyzed Mad2 Structural Transition and Spindle Checkpoint Silencing” by Brulotte et al. addresses most of my major concerns. As my earlier review made clear, I consider the work important and worthy of publication - especially the results in Figure 7. I do have three remaining important points that I hope the authors will address before publication.

1) The authors continue to refer to their structure of TRIP13 as “monomeric”, when it is nothing of the sort. In fact, the structure is of a six-fold symmetric helical polymer whose inter-subunit interfaces closely match those of the earlier-solved structure of *C. elegans* PCH-2. Think of it this way: if the authors were to index these crystals in space group P1 instead of P65, such that there were then six subunits in the asymmetric unit in a six-fold symmetric helical filament (as they are arranged now), would the authors continue to refer to this as a “monomer”? If not, I suggest they change this to more accurately reflect what their structure is showing them, without regard to crystallographic symmetry.

2) The authors go to great lengths to invalidate the cysteine cross-linking data from the earlier Ye...Corbett paper that purport to show direct engagement of the MAD2 N-terminus by the PCH-2 pore loops. Without commenting on the earlier results directly, I would note that the authors appear to have missed a couple of important points about that assay. The authors are correct that the cross-linking experiments used a TRIP13 W221C mutant, and I have no doubt that they are correct that TRIP13 W221C hexamers are inactive, much like the W221A mutant hexamers. But, Ye...Corbett used a 1:8 ratio of W221C to wild-type TRIP13 in their cross-linking assays, such that only one subunit of each hexamer would be expected to contain the mutation. Work with other AAA+ protein remodelers has shown that mutating a single pore loop is not enough to inactivate the remodeling activity. Indeed, a similar cross-linking assay was performed with Vps4, again with a cysteine mutation of a key pore-loop tryptophan residue, introduced at a low ratio (Yang...Hurley NSMB 2015). In a related point, the TRIP13 used in the cysteine cross-linking assay was clearly actively remodeling its substrate, as demonstrated by the need for ATP, as opposed to ATP- γ -S, for cross-linking to internally-placed cysteine residues in MAD2. The authors should not so strongly discount these earlier assays without addressing these points.

3) The authors introduce a delta-N10 mutant of MAD2 to further address the question of whether the MAD2 N-terminus is required for TRIP13-mediated remodeling. The authors show in Figure 7 that this MAD2 mutant is somewhat defective in TRIP13 remodeling, but importantly they fail

to note in the manuscript that the delta-N 10 mutant is in fact longer than wild-type MAD2, due to the presence of a 12-residue N-terminal tag. If the length of the MAD2 N-terminus, rather than its exact sequence, is what is important for TRIP13-mediated remodeling, then this delta-N10 MAD2 is not the right reagent. In any case, this point should be noted in the manuscript.

Reviewer #2 (Remarks to the Author):

The paper describes a collection of results from bio-physical and –chemical experiments that elucidate the mechanism of closed-to-open state transition of Mad2 catalyzed by TRIP13. This is then related to the role of TRIP13 in spindle checkpoint silencing. Given the high significance of correct spindle checkpoint silencing in chromosome segregation and therefore also in many diseases, detailed understanding of the mechanism is important.

In detail, the findings are:

- 1) TRIP13 + p31comet catalyze Mad2 opening without disrupting its core fold, driven by ATP hydrolysis.
- 2) Determination of the X-ray structure bound to ADP (rather than ATP).
- 3) TRIP13 + p31comet reverse inhibition of APC/C via Mad2.
- 4) Assays identify the role of specific TRIP13 residues in ATP hydrolysis or binding to p31comet.

Taken together, the paper provides a deepened and interesting insight into the investigated mechanisms. In particular, findings 3 and 4 are valuable.

Response: We thank the reviewer for the overall positive comments about our work and for pointing out the importance of our findings.

On the other hand, the NMR investigations are somewhat underwhelming. NMR spectroscopy demonstrates that TRIP13 indeed catalyzes the Mad2 rearrangement, which has already been shown previously, see ref. 69. The only new finding is that the unfolding takes place without disruption of the core of Mad2. This finding is convincing thanks to tracing H/D exchange. It appears that the authors did not expect this. Why so? I would have been surprised if complete unfolding had been involved. Is it because ClpX involves complete unfolding? What molecule does ClpX unfold? Is it comparable to Mad2? This should be discussed.

Response: We agree with the reviewer that our demonstration of TRIP13-catalyzed Mad2 conformational change is not entirely novel, and we did not claim that. The original ref. 69 (now ref. 55) showed that TRIP13 catalyzes Mad2 structural transition by using anion exchange chromatography. We wish to point out that NMR spectroscopy is a more definitive method to demonstrate the conformational change of Mad2. In fact, it was not trivial to find the right

conditions to demonstrate this by NMR. In any case, the fact that multiple methods confirm the same finding will be reassuring to the field. With the H/D exchange experiments, we were simply trying to differentiate two possible models of TRIP13-catalyzed Mad2 conformational change: global unfolding versus local unfolding. Before we performed the experiment, we did not favor or disfavor either model. We did not find the local unfolding result particularly unexpected. We have removed the word “Strikingly” from the text to avoid any confusion. The substrates of ClpX are bacterial proteins, which are not structurally related to Mad2. We have added a statement in the Discussion to clarify this.

There is not really a structural picture of the mechanism emerging (the authors promise “key insight into the mechanism of TRIP13-catalyzed conformational change of Mad2”). It seems relatively easy to collect much more information from NMR. For example, the X-ray attempt to solve the structure of the complex failed. Why not checking the involved residues by NMR such that a crude complex model could be provided nevertheless? Simple chemical shift mapping of the Mad2 peaks upon TRIP13+p31comet titration would show where the binding hot spots are. As the reaction takes hours it would also be interesting to observe the changes of the peaks over time in HSQC spectra. Therefore, the new insights from a structural point of view are somewhat incremental. The NMR investigation stopped where it would turn interesting.

Response: We agree that the crystal structure of the TRIP13-p31comet-Mad2 complex would have been more informative. On the other hand, we feel that our biophysical and biochemical experiments, coupled with extensive mutagenesis, have provided key insight into the mechanism of TRIP13. Note that we have refrained from claiming structural insight. The NMR experiments suggested by the reviewer are, unfortunately, not technically feasible at present. To map chemical shift perturbations of Mad2, we would need to add p31comet and TRIP13 at stoichiometric amounts (as opposed to the 1:33 or 1:325 molar ratios used in the catalysis experiment). We have attempted to acquire the HSQC spectrum (the TROSY version) of the Mad2-p31comet complex, with only Mad2 labeled with ^{15}N . As shown in Figure-to-reviewer, virtually all Mad2 peaks disappeared when it is bound to p31comet. Therefore, the relatively large size of the Mad2-p31comet complex (50 kD) and quite possibly exchange broadening prohibit us from acquiring usable NMR spectra on that complex. The size of the Mad2-p31comet-TRIP13 complex is about 300 kD. NMR studies of such a large complex would be even more challenging. In addition, we cannot concentrate the Mad2-p31comet-TRIP13 complex to concentrations higher than $10\ \mu\text{M}$. Therefore, it is impractical to study the complex by NMR spectroscopy. We are, however, attempting to determine the structure of the complex with cryo-electron microscopy. This effort is beyond the scope of the current paper. Unfortunately, the paper is already too long (the journal asks that the main text be under 5,000 words). There is no space to add this discussion to the text. We hope that the reviewer would understand.

Figure-to-reviewer. (a) Coomassie-stained SDS-PAGE gel of the purified recombinant Mad2-p31comet complex that contains ^{15}N -labeled C-Mad2R133A and unlabeled $\Delta\text{N35-p31comet}$. (b) Overlay of the ^1H - ^{15}N TROSY-HSQC spectra of C-Mad2R133A alone (in red) and C-Mad2R133A- $\Delta\text{N35-p31comet}$ complex (in blue). The sample concentration is $140\ \mu\text{M}$ of proteins. Each spectrum lasting 24 hours was acquired on an 800 MHz NMR spectrometer.

I also wonder if a NMR analysis of the TRIP13 is possible. Could TRIP13 be studied in its monomeric form (X-ray suggests so)? If so, chemical shift analysis could be repeated on TRIP13 and extend the insight considerably. It may even be possible to conduct CPMG experiments to substantiate the repeatedly mentioned conformational exchange dynamics. (However, this would be clearly beyond the scope of this paper.)

Response: Even though the structure of TRIP13 E253A in the crystal is monomeric, the same protein is hexameric in solution. We currently do not understand why this is. In any case, we do not have a TRIP13 mutant that is monomeric in solution for us to attempt NMR analysis. Moreover, lessons learned from a monomeric mutant of TRIP13 may not be relevant to the functional state of the protein, which is hexameric. If technically feasible, CPMG experiments on the TRIP13 monomer and hexamer might be informative, but this would require considerable effort, including the development of novel methyl labeling schemes and ways to improve the solubility of the protein. As stated by the reviewer, this is clearly beyond the scope of this paper.

The following points should also be addressed:

-The NMR proof of the Mad2 conversion upon addition of TRIP13 is based on a HSQC spectrum before and after addition of TRIP13, which exhibits the peaks expected for the C- or O-conformation. What samples were used for the reference spectra (magenta and blue in figure 1b)? Apparently, there is another way to induce the conformational switch.

Response: As we have shown previously (ref. 13), Mad2 expressed in bacteria has two conformers, which can be separated by anion exchange chromatography: C-Mad2 elutes at higher salt as compared to O-Mad2. This allows us to obtain both forms of Mad2 and acquire reference spectra. C-Mad2 is more thermodynamically stable than O-Mad2. At 30°C , O-Mad2 spontaneously converts to C-Mad2, with a half-life of 9 hours. At low temperature (4°C), this conversion is extremely slow. In the absence of p31comet, TRIP13, and ATP, C-Mad2 does not convert back to O-Mad2.

-What is rational behind the choice of the mutants? Since the complex structure is not known, the optimal choice is not that obvious.

Response: We targeted all TRIP13 residues conserved from yeast to man that are surface exposed or partially exposed in the hexameric C. elegans PCH-2 structure. Our rationale is that

only surface exposed residues have the chance to form direct contact with p31comet-Mad2, and functionally important residues are often conserved. Not all mutants are presented in the study. We have now clarified this point in the revised manuscript.

-The pore loop should be indicated in figure 3, which would help to follow the extensive discussion on page 9.

Response: This is a great suggestion. We have now labeled the pore loop of PCH-2 in Fig. 3c. The pore loop is missing in the structure of human TRIP13.

-Why is the activity in figures S2 c) and d) not the same for the wild type?

Response: The basal level ATPase activity of TRIP13 is low, and is variable among different TRIP13 preparations. Supplementary Fig. 2c was performed with an earlier preparation of TRIP13, which was less active.

-An interesting finding is that the G320A mutation has a major impact. While it is not unreasonable to assume that the mutation affects dynamics, I would tone down that claim in the title on page 13. It would also be interesting to see a close-up of the hinge loop including side chains. It is not clear if residue 320 is located at the hinge or in the middle of the loop.

Response: We agree with the reviewer. We do not have direct evidence that the hinge actually impacts TRIP13 conformational dynamics. We have changed the title of this section to “The hinge impacts TRIP13 activity and oligomerization”, which better reflects our results. We have now shown G320 in sticks in Fig. 5d and 6c. Because G320 has no side chain, we have shown the main chain. G320 is located in the middle of this loop. The loop is very short, and we refer to the entire loop as the hinge.

-It is not clear to me how the ATPase activity is coupled to the transition. Could the authors add one or two phrases explaining this?

Response: We do not know exactly how the ATPase activity is coupled to the structural transition of Mad2. As outlined in Fig. 8a, we propose that the C-terminus of Mad2 engages with the pore loop of TRIP13. Movement of the pore loop driven by ATP hydrolysis tugs on this region of Mad2. The C-terminal region of Mad2 may be threaded into the central pore of TRIP13. Alternatively, the translocation of the Mad2 C-terminus driven by the pore loop may occur exclusively at the top face of TRIP13. In either case, the ATP-driven pore loop movement causes the local unfolding of the Mad2 C-terminal region and its conformational transition to O-Mad2. We have made this point more clearly in the revised manuscript.

Review of rebuttal:

The points I have raised are addressed to my satisfaction. Due to the importance and quality of this study, I recommend the publication of this article in Nature Communications.

One small thing that the authors forgot to address:

Figure captions 1b and c: As far as I understand, C-Mad2 and O-Mad2 are also ^{15}N labeled, so it is confusing to name the construct that is titrated “ ^{15}N -Mad2”. Or are C-Mad2 and O-Mad2 taken at natural abundance? This should be clarified in the caption.

Reviewer #1:

1) The authors continue to refer to their structure of TRIP13 as “monomeric”, when it is nothing of the sort. In fact, the structure is of a six-fold symmetric helical polymer whose inter-subunit interfaces closely match those of the earlier-solved structure of C. elegans PCH-2. Think of it this way: if the authors were to index these crystals in space group P1 instead of P65, such that there were then six subunits in the asymmetric unit in a six-fold symmetric helical filament (as they are arranged now), would the authors continue to refer to this as a “monomer”? If not, I suggest they change this to more accurately reflect what their structure is showing them, without regard to crystallographic symmetry.

Response: We have removed explicit statements that human TRIP13 E253A is monomeric in the crystals, and modified the text to state that the human TRIP13 structure does not capture the closed hexamer conformation of the ATPase.

2) The authors go to great lengths to invalidate the cysteine cross-linking data from the earlier Ye... Corbett paper that purport to show direct engagement of the MAD2 N-terminus by the PCH-2 pore loops. Without commenting on the earlier results directly, I would note that the authors appear to have missed a couple of important points about that assay. The authors are correct that the cross-linking experiments used a TRIP13 W221C mutant, and I have no doubt that they are correct that TRIP13 W221C hexamers are inactive, much like the W221A mutant hexamers. But, Ye... Corbett used a 1:8 ratio of W221C to wild-type TRIP13 in their cross-linking assays, such that only one subunit of each hexamer would be expected to contain the mutation. Work with other AAA+ protein remodelers has shown that mutating a single pore loop is not enough to inactivate the remodeling activity. Indeed, a similar cross-linking assay was performed with Vps4, again with a cysteine mutation of a key pore-loop tryptophan residue, introduced at a low ratio (Yang... Hurley NSMB 2015). In a related point, the TRIP13 used in the cysteine cross-linking assay was clearly actively remodeling its substrate, as demonstrated by the need for ATP, as opposed to ATP-gamma-S, for cross-linking to internally-placed cysteine residues in MAD2. The authors should not so strongly discount these earlier assays without addressing these points.

Response: We thank the reviewer for pointing out that the previous study used W221C as a sub-stoichiometric mixture with WT. We have now discussed this point more clearly in the revised paper. Although Baker, Sauer and others have shown (by using covalent trimers) that mutations of one or two copies of the pore loop in other AAA+ ATPases (e.g. ClpX) do not abolish the unfolding activities, these mutations even at low copy numbers are not completely functional and weaken the processivity of the enzyme. It is thus possible that the mutated pore loop even in the presence of WT copies in the same hexamer might not make the correct contact with the substrate. Crosslinking data with the use of this system are complicated by this caveat. This caveat applies to other published studies on Vps4 and ClpX. Furthermore, the cysteines in Mad2 used by the Corbett study are solvent accessible, and can be crosslinked to other partners without Mad2 unfolding. Their results would have been much stronger, had they installed a cysteine that is completely buried in the hydrophobic core of Mad2. One possible explanation of the ATP-dependence of the observed crosslinking is that ATP hydrolysis leads to transient dissociation of the TRIP13 hexamer, making the mutated pore loop more accessible for crosslinking. With this

said, we agree with the reviewer that we need to be more cautious. We have toned down our criticism of the previous work, and simply mentioned the caveat of their experiment.

3) The authors introduce a delta-N10 mutant of MAD2 to further address the question of whether the MAD2 N-terminus is required for TRIP13-mediated remodeling. The authors show in Figure 7 that this MAD2 mutant is somewhat defective in TRIP13 remodeling, but importantly they fail to note in the manuscript that the delta-N 10 mutant is in fact longer than wild-type MAD2, due to the presence of a 12-residue N-terminal tag. If the length of the MAD2 N-terminus, rather than its exact sequence, is what is important for TRIP13-mediated remodeling, then this delta-N10 MAD2 is not the right reagent. In any case, this point should be noted in the manuscript.

Response: We agree with the reviewer that, though unlikely, there is a formal possibility that the extra two residues in length in the Δ N10-Mad2 protein, not the change of amino acid sequence, accounts for the weakened binding to TRIP13. We have mentioned this possibility in the revised text.

Reviewer #2:

One small thing that the authors forgot to address:

Figure captions 1b and c: As far as I understand, C-Mad2 and O-Mad2 are also ^{15}N labeled, so it is confusing to name the construct that is titrated “ ^{15}N -Mad2”. Or are C-Mad2 and O-Mad2 taken at natural abundance? This should be clarified in the caption.

Response: The reviewer is correct. The C-Mad2 and O-Mad2 are both ^{15}N -labeled. We have changed the captions of Fig. 1b,c to clarify this point.

Reviewers' Comments:

Reviewer #1 (Remarks to the Author):

I'm satisfied with the changes the authors have made, and support publication.

Reviewer #1 (Remarks to the Author):

I'm satisfied with the changes the authors have made, and support publication.

Response: We thank the reviewer for his support.